# A modular architecture for trial-by-trial learning of redundant muscle activity patterns in novel sensorimotor tasks

Lucas Rebelo Dal'Bello[1]*, Denise Jennifer Berger[1,2], Daniele Borzelli[3], Etienne Burdet[4], Andrea d'Avella[1,5]

**1** Laboratory of Neuromotor Physiology, Fondazione Santa Lucia IRCCS, Rome, Italy, **2** Department of Systems Medicine and Centre of Space Bio-medicine, University of Rome Tor Vergata, Rome, Italy, **3** Laboratory of Physiology, Department of Translational Medicine, Università del Piemonte Orientale, Novara, Italy, **4** Department of Bioengineering, Imperial College of Science, Technology and Medicine, London, United Kingdom, **5** Department of Biology, University of Rome Tor Vergata, Rome, Italy

* l.dalbello@hsantalucia.it

## Abstract

The coordination of the multiple degrees-of-freedom of the human body may be simplified by muscle synergies, motor modules which can be flexibly combined to achieve various goals. Studies investigating adaptation to novel relationships between muscle activity and task outcomes found that altering the recruitment of such modules is faster than the learning of their structures *de novo*. However, how learning new synergy recruitments or new synergy structures may occur remains unclear. While trial-by-trial learning of novel sensorimotor tasks has been successfully modeled at the level of task variables, few models accounted for the redundancy of the motor system, particularly at the muscular level. However, these models either did not consider a modular architecture of the motor system, or assumed *a priori* knowledge of the sensorimotor task. Here, we present a computational model for the generation of redundant muscle activity where explicitly defined modules, implemented as spatial muscle synergies, can be updated together with their recruitment coefficients through an error-based learning process dependent on a forward model of the sensorimotor task, which is not assumed to be known *a priori*. Our model can qualitatively reproduce the experimental observations of slower learning and larger changes in the structure of the muscle activity under sensorimotor tasks that require the learning of novel patterns of muscle activity, providing further insights into the modular organization of the human motor system.

## Author summary

It has been proposed that muscles are recruited in modules, called muscle synergies, rather than on a muscle-by-muscle basis. This modular organization

which permits unrestricted use, distribution, and reproduction in any medium, provided the original author and source are credited.

**Data availability statement:** The GitHub repository with the code used in our simulations has been archived in Zenodo, and is accessible through the following DOI: https://doi.org/10.5281/zenodo.19131019 All other relevant data are in the manuscript and its supporting information files.

**Funding:** LD gratefully acknowledges the funding provided by the Italian Ministry of Health (IRCCS Fondazione Santa Lucia, Ricerca corrente; GR-2019-12370271). DJB gratefully acknowledges the funding provided by the #NEXTGENERATIONEU (NGEU) National Recovery and Resilience Plan (NRRP), project MNESYS (PE0000006)—A Multiscale integrated approach to the study of the nervous system in health and disease (DN. 1553 11.10.2022). DB gratefully acknowledges the Università del Piemonte Orientale for the support provided through the Starting Kit 2025. AD gratefully acknowledges the funding provided by the project "HARIA - Human-Robot Sensorimotor Augmentation - Wearable Sensorimotor Interfaces and Supernumerary Robotic Limbs for Humans with Upper-limb Disabilities" (EU Horizon Europe, GA No. 101070292, https://clem.diism.unisi.it/~haria/) and by the Italian Ministry of Research (PRIN 2022YXLNR7). The funders had no role in study design, data collection and analysis, decision to publish, or preparation of the manuscript.

**Competing interests:** The authors have declared that no competing interests exist.

was shown to affect the learning of novel tasks, where simulated remapping of the forces generated by the muscles ("virtual surgeries") that make the synergies ineffective is more difficult to learn. Previous models of trial-by-trial learning in a modular architecture have assumed prior knowledge of the motor task (in particular, how to modify the muscles/ synergies recruitment given an error), which might not be the case for tasks in which novel patterns of muscle activity are required. In contrast, models which assumed no prior knowledge of the task have not investigated the role of modularity. Here, we propose a computational model of the trial-by-trial adaptation of muscle synergies, their recruitment, and the concurrent learning of the internal model of the musculoskeletal system responsible for error correction. Our results show that this model replicates experimental observations of slower learning and larger changes in the structure of the muscle activity in sensorimotor tasks that require the learning of novel patterns of muscle activity, providing insight into learning-related changes of muscle activity in novel sensorimotor tasks.

## 1 Introduction

During movement planning and execution, the central nervous system needs to coordinate the activity of a large and redundant set of muscles acting on multiple joints [1]. It has been suggested that the problem of coordinating the multiple degrees-of-freedom of the human body is simplified by grouping multiple muscles into a reduced number of motor modules, often called muscle synergies, which can be flexibly combined to generate a large repertoire of movements [2]. The modularity of the motor system has been investigated through the decomposition of the muscle activity patterns measured either during voluntary movements or in response to cutaneous or spinal cord electrical stimulation. The key observation is that the electromyography (EMG) signals recorded from many muscles can be accurately reconstructed by the combination of a small number of modules that generalize across multiple contexts and movements [3–6]. However, this low-dimensionality observed in the muscle activity patterns may be due to task and biomechanical constraints rather than to a modular organization of the motor system [7].

Further support for the modular organization of the motor system has come from testing specific predictions of how modularity would affect a novel motor learning task [8]. In this task, participants generated a virtual force to control a cursor using their isometric muscle activity measured with EMG and adapted to two different perturbations of the mapping of muscle activity to virtual force, or "virtual surgeries". These perturbations consisted of changes to the directions of the virtual forces generated by the contraction of each muscle, akin to the effect of a complex tendon-transfer surgery involving multiple muscles. The two types of surgeries differed in how effective the muscle synergies identified before the surgery were in generating a virtual force. After compatible surgeries, the muscle synergies still spanned the entire force space, while incompatible surgeries reduced the space of forces that can be spanned by the

muscle synergies. However, the force space was still spanned by the individual muscles under both types of surgeries, so that a modular organization of the motor system predicted a difference in the adaptation rate to the two types of surgeries, while a non-modular organization predicted no difference. The observation that learning was slower during incompatible virtual surgeries provided stronger evidence for the modular organization of the motor system.

In addition to their role in movement generation, muscle synergies also play a role in motor adaptation and motor learning. During visuomotor rotations, it has been shown that the directional tuning of the synergy recruitment is a possible adaptation strategy employed by the central nervous system [9]. As previously mentioned, in studies that investigated the adaptation to novel relationships between muscle activity and the resulting end-effector force using virtual surgeries, it was shown that surgeries which were compatible with the original synergies (requiring only a change in the synergy directional tuning) were learned faster than surgeries which were incompatible (requiring the usage of novel muscle activity patterns), suggesting that the altering of the synergy recruitment coefficients is faster than the learning *de novo* of new synergy structures [8, 10]. These results suggest that the modularity of the motor system influences its adaptation to perturbations and its learning of novel patterns of activity, which may be relevant to neurorehabilitation [11].

While the adaptation of hand reaching movements under externally-induced visuomotor errors has been extensively studied and modeled [12], relatively few studies have focused on changes at the highly-redundant muscular level. The adaptation of feedforward muscle activity in response to feedback error under force-field perturbations has been modeled by a "V-shaped" learning function, explaining the regulation of reciprocal- and co-activation in stable and unstable conditions [13]. Similar to changes to the directional tuning of muscle synergies under visuomotor rotations [9], changes in muscles' tuning curves under force-field perturbations have also been reported [14]. When accounting for the modularity of the motor system, the adaptation to force-field perturbations [15] and to virtual surgeries incompatible with the original synergies [8] produced muscle activity patterns poorly explained by the original synergies. Such a change in the muscle activity patterns was shown not to interfere with performance in the baseline task, and was persistent even during a subsequent exposure to a compatible surgery, suggesting an increase in the exploration of muscular null-space activity patterns [10]. Despite these important advances, the error-based learning mechanisms governing changes at the muscular level, including learning of novel muscle activity patterns, are still unclear.

Learning novel patterns of muscle activity, and not simply reusing already known patterns, is linked to the acquisition of new motor skills, a process also known as *de novo* learning [16]. While the exact mechanisms of *de novo* learning are still unclear, it has been argued that it requires the learning of an internal model responsible for error corrections, which is used to guide the update of the controller for the motor task [17–19]. A few studies have investigated the learning of such error-correcting internal models simultaneously through a trial-by-trial update of a feedforward controller [20], with one study focusing on the role of motor exploration in the learning process [21]. However, these studies did not account for the possible modularity of the motor system. In contrast, studies that include modularity assumed that the error-correcting internal model was known a *priori*, which might not be the case in *de novo* learning [22,23]. Also, one of these studies [23] did not investigate a possible update of the synergies' structure, which has been observed during training under incompatible virtual surgeries [8,10].

Here, we introduce a model of trial-by-trial, error-based learning of motor commands in a modular control architecture that addresses mechanisms of learning in a redundant motor system. Feedforward muscle activity patterns during isometric force-reaching tasks are generated using explicitly defined muscle synergies, whereas novel motor commands are learned through trial-by-trial updates of both the synergies and their recruitment coefficients at distinct learning rates. These updates are driven by the backpropagation of the difference between the intended and executed force through a forward model of the isometric task. Importantly, this forward model is also updated over time, reflecting learned changes in the sensorimotor system or in the task environment. We show that the model can qualitatively reproduce results from multidirectional isometric force generation tasks under different perturbations, including visuomotor rotations and virtual surgeries that rearrange the directions of the forces produced by individual muscles. We analyze the model's predictions

 

in terms of changes in the force error and muscle activity structure across different perturbations, and under conditions with or without updates to synergy recruitment, synergy structure, and forward model. We also compare our simulations with available experimental data and discuss potential explanations for some of the discrepancies.

## 2 Results

### 2.1 Computational model of trial-by-trial adaptation

We model the trial-by-trial adaptation of the muscle activity patterns during the generation of isometric force at the hand (Fig 1) in a redundant musculoskeletal system that can be exposed to perturbations such as visuomotor rotations and

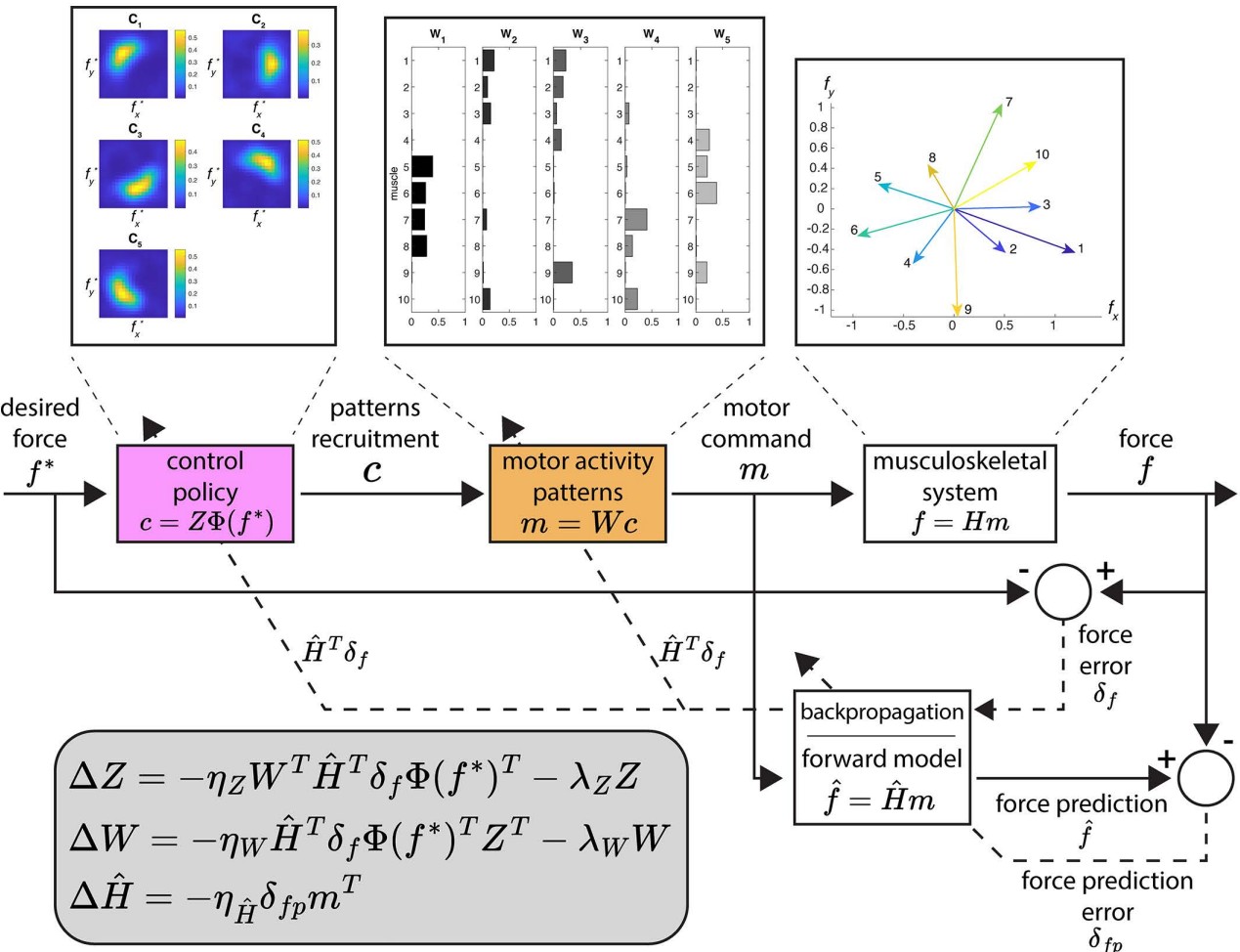

**Fig 1. Computational model of trial-by-trial generation and update of redundant muscle activity for isometric force control.** The model maps a desired force into a recruitment of muscle synergies using a radial basis function-based control policy. The combination of the spatial muscle synergies results in a muscle activity pattern executed through the musculoskeletal system. Both the control policy and the synergy structure can be updated by backpropagating the force error through a forward model of the musculoskeletal system. This forward model can also be updated using the force prediction error. Solid lines indicate the "forward" flow of information in the model, from a desired force to the generation of a motor command and the force execution (by the musculoskeletal system) and from a motor command to a force prediction (by the forward model). Dashed lines indicate the "backward" flow of information, from errors (force error and force prediction error) to their use in updating the model components (control policy and muscle synergies; and forward model, respectively). Insets at the top represent the control policy's recruitment of each synergy given a desired force, the synergies' recruitment of each muscle, and the forces executed by each muscle in the musculoskeletal system.

'virtual surgeries' [8]. For this purpose, we assume that the generation of the redundant muscle activity occurs through the recruitment of modules representing spatial muscle synergies: a muscle pattern $\boldsymbol{m}$, i.e., a non-negative $M$-dimensional vector of activation of a set of $M$ muscles, is generated by the linear combination of $N$ muscle synergies, each a non-negative $M$-dimensional vector:

$$\boldsymbol{m} = \boldsymbol{Wc} \tag{1}$$

where $\boldsymbol{W}$ is an $M$ x $N$ matrix with the synergy vectors as columns, and $\boldsymbol{c}$ is a $N$-dimensional synergy combination vector. The activation of each muscle generates a specific isometric force at the virtual hand, which we approximate as a linear function of activation, and the joint activation of all muscles results in an end-effector force $\boldsymbol{f}$ ($D$-dimensional vector):

$$\boldsymbol{f} = \boldsymbol{Hm} \tag{2}$$

where $\boldsymbol{H}$ is the $D$ x $M$ matrix that linearly maps the muscle activation pattern into a $D$-dimensional force.

The synergies are recruited according to a control policy that maps a desired force $\boldsymbol{f}^*$ into synergy recruitment coefficients. We implement this control policy using radial basis functions, which have been shown to reproduce well the generalization patterns observed in motor adaptation experiments and have been extensively used in fitting primitives of internal models for movement planning [24–26]. We implement this radial basis function-based control policy with:

$$\boldsymbol{c}\left(\boldsymbol{f}^*\right) = \boldsymbol{Z}\Phi\left(\boldsymbol{f}^*\right) \tag{3}$$

where $\boldsymbol{Z}$ is a $N$ x $N_\phi$ matrix of non-negative combination coefficients and $\Phi\left(\boldsymbol{f}^*\right)$ is a $N_\phi$-dimensional vector of Gaussian basis functions activations, with centers spread in a region of the $D$-dimensional force space. The generation of a muscle activity pattern can then be written as:

$$\boldsymbol{m}\left(\boldsymbol{f}^*\right) = \boldsymbol{WZ}\Phi\left(\boldsymbol{f}^*\right). \tag{4}$$

In our simulations, we also add signal-dependent, random motor noise to the muscle activity obtained from the model, sampled from a multivariate Gaussian distribution with the variance of each muscle scaled quadratically with that muscle's activation [27] and with zero covariance between different muscles. Signal-dependent noise has been observed experimentally in motor unit firing rates [28,29], in EMG [30], and in isometric force production [31]; therefore, including it improves the biological realism of the model. After adding motor noise, any negative muscle activity values are set to zero to ensure non-negativity.

Previous models of feedforward generation of redundant muscle activity [22,23] assume that the environment—i.e., the relationship between muscle activation and the resulting end-effector force, accounting for the musculoskeletal system and external perturbations—is either known *a priori* or learned instantaneously upon perturbation. In contrast, our model assumes that the environment is learned gradually through practice. It is represented by an internal estimate $\hat{\boldsymbol{H}}$ of the muscle activity-to-force matrix $\boldsymbol{H}$ that serves as an internal forward model, allowing the prediction of the generated forces $\hat{\boldsymbol{f}}$ from a given a muscle activity pattern $\boldsymbol{m}$:

$$\hat{\boldsymbol{f}} = \hat{\boldsymbol{H}}\boldsymbol{m}. \tag{5}$$

A control policy is learned by updating the synergy matrix $\boldsymbol{W}$ and the combination coefficients matrix $\boldsymbol{Z}$ through minimization of the squared norm of the force error $\delta_f = \boldsymbol{f} - \boldsymbol{f}^*$, defined as the difference between the executed force $\boldsymbol{f}$ and the desired force $\boldsymbol{f}^*$. The matrices $\boldsymbol{W}$ and $\boldsymbol{Z}$ are then updated based on the gradient of the squared norm of the error, $\frac{1}{2}\|\delta_f\|^2$, with respect to each matrix (derivations of the update equations are provided in S1 Text). We also add a regularization

term to each matrix update to reduce co-contraction of muscles and co-recruitment of synergies, i.e., effort. The regularization is defined as a minimization of the entry-wise squared norm of the control policy and synergy matrices. To ensure the non-negativity of the muscle activity, any element of the matrices $\boldsymbol{W}$ and $\boldsymbol{Z}$ that are negative after an update are set to zero.

For the synergy structure matrix $\boldsymbol{W}$, the update $\triangle\boldsymbol{W}$ after a trial with a desired force $\boldsymbol{f}^*$ resulting in an error $\delta_f$ is:

$$\triangle\boldsymbol{W} = -\eta_W\hat{\boldsymbol{H}}^T\delta_f\Phi\left(\boldsymbol{f}^*\right)^T\boldsymbol{Z}^T - \lambda_W\boldsymbol{W} \qquad (6)$$

where $\eta_W$ is a scalar learning rate and $\lambda_W$ is a scalar regularization weight. This contrasts with previous models of the generation of redundant motor activity that either did not consider a modular controller [20,21], or assumed fixed synergies [23].

The update $\triangle\boldsymbol{Z}$ of the combination coefficients matrix $\boldsymbol{Z}$ is:

$$\triangle\boldsymbol{Z} = -\eta_Z\boldsymbol{W}^T\hat{\boldsymbol{H}}^T\delta_f\Phi\left(\boldsymbol{f}^*\right)^T - \lambda_Z\boldsymbol{Z} \qquad (7)$$

where $\eta_Z$ is a scalar learning rate and $\lambda_Z$ is a scalar regularization weight. It is important to note that both update equations contain the matrix $\hat{\boldsymbol{H}}^T$ of the internal forward model of the environment, which here serves as an estimate of the "sensitivity derivative" $\frac{\partial\boldsymbol{f}}{\partial\boldsymbol{m}}$ of the environment [32]. When calculating the gradient of the squared norm of the error $\frac{1}{2}\|\delta_f\|^2$, both update equations would contain the matrix $\boldsymbol{H}^T$ of the real environment which is not *a priori* known. Since we assume that the learner uses an internal estimate of the environment, the matrix $\hat{\boldsymbol{H}}^T$ is used instead in the update equations [17].

Equations 6 and 7 suggest that, in case of a large discrepancy between the environment $\boldsymbol{H}$ and the internal model of the environment $\hat{\boldsymbol{H}}$, the update of matrices $\boldsymbol{W}$ and $\boldsymbol{Z}$ of the controller will not necessarily minimize the norm of the error $\delta_f$. Indeed, it has been shown that a sufficient condition for the internal model of the environment $\hat{\boldsymbol{H}}$ to decrease the squared norm of the error during the update of the model is that $\delta_f^T\boldsymbol{H}\hat{\boldsymbol{H}}^T\delta_f > 0$, that is, the vector $\hat{\boldsymbol{H}}^T\delta_f$ (present in both update equations) has an angle of less than 90° with the vector $\boldsymbol{H}^T\delta_f$, which uses the original environment matrix $\boldsymbol{H}$ [33].

We update the internal forward model $\hat{\boldsymbol{H}}$ by using the gradient of the norm of the prediction error $\delta_{fp} = \hat{\boldsymbol{f}} - \boldsymbol{f}$ between the force predicted by the forward model given a muscle activity $\boldsymbol{m}$ and the actual generated force for the same muscle activity. The update $\triangle\hat{\boldsymbol{H}}$ can be written as:

$$\triangle\hat{\boldsymbol{H}} = -\eta_{\hat{H}}\delta_{fp}\boldsymbol{m}^T \qquad (8)$$

where $\eta_{\hat{H}}$ is a scalar learning rate.

## 2.2 Qualitative reproduction of experimental results

Fig 2 illustrates the three types of perturbations tested in our model and their effect on the forces generated by the model's initial muscle synergies (Fig 2A). Visuomotor rotations, which rotate the executed force around the task origin, have been extensively studied in the motor control literature [12,22,34,35], and are used here to benchmark the model's learning process. Compatible and incompatible virtual surgeries modify the relationship between the initial muscle synergies of the model and the forces they produce. In compatible surgeries, the synergies still span all directions in the force space, so reaching each target requires only adjustments in their relative recruitment. In contrast, incompatible surgeries cause the initial synergies to no longer span the full force space, requiring modifications to the synergies to reach certain force targets. Human participants adapt faster to compatible than to incompatible surgeries (Fig 2B), supporting the idea that the motor system is organized in a modular architecture, with changes in synergy structure occurring more slowly than adjustments in their relative synergy activations. Details of the virtual surgery computation are provided in the Methods section.

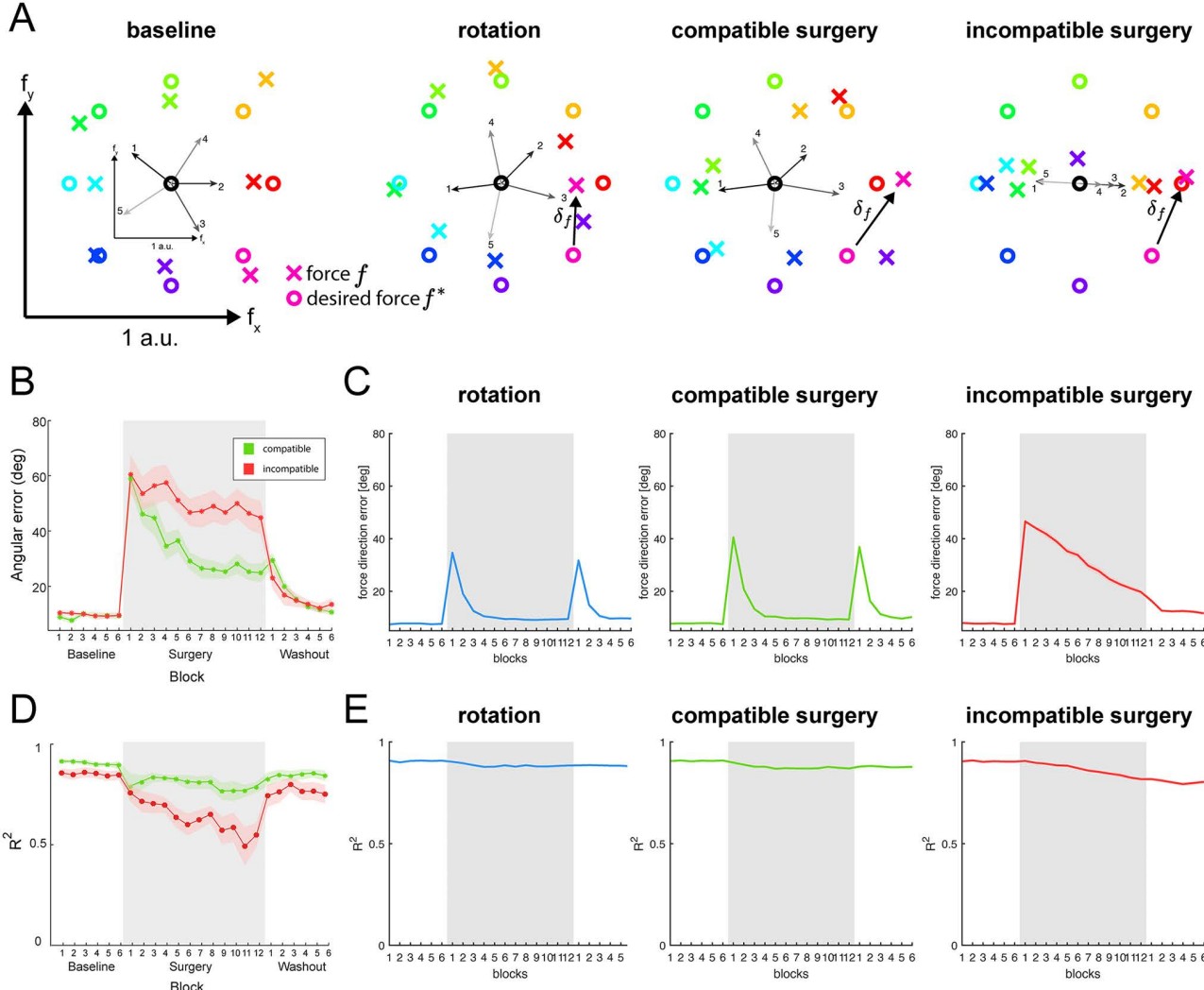

**Fig 2. Simulated perturbations and qualitative reproduction of experimental results. (A)** Diagrams show the force generated by the computational model (*colored crosses*) to match each of 8 targets (*colored circles*) at baseline and at the onset of each of three types of perturbation (rotation, compatible surgery, incompatible surgery), as well as the effect of the perturbations on the forces generated by the five muscle synergies (*black arrows*, numbered and shown in a smaller scale). During baseline, all force targets can be reached adequately (small errors are mostly due to motor noise). During a visuomotor rotation, the generated forces are rotated around the center of the task space, resulting in force errors. During a compatible surgery, the synergy forces are altered but still span the entire force space. In contrast, during an incompatible surgery, the baseline muscle synergies do not span the entire force space, and the generated forces are initially all aligned along a single axis. **(B)** Initial angle error of human participants performing under compatible and incompatible virtual surgeries (reproduced from [8]), showing a slower decrease of the error under incompatible surgeries (averaged across eight participants). **(C)** Force direction error of the simulated model (analogous to the initial angle error in data from experiments with human participants, see Methods) during adaptation to the three perturbations (average across sixteen initializations of the model). **(D)** Reconstruction quality ($R^2$) of the muscle activity using the original synergies from human participants performing under compatible and incompatible virtual surgeries (reproduced from [8]), showing a larger decrease in the reconstruction quality under incompatible surgeries (averaged across eight participants). **(E)** Reconstruction quality ($R^2$) of the muscle activity using the original synergies from the simulated model during adaptation to the three perturbations (average across sixteen initializations of the model).

Fig 2C shows the force direction error over the course of training in simulations involving the three different perturbations, with all model elements being updated. During the visuomotor rotation, the force direction error decreases substantially within a few cycles (each cycle consists of eight trials, with each trial involving the generation of an isometric force

towards one of the eight different targets), consistent with experimental findings on adaptation under visuomotor rotations [34,35]. Importantly, the decrease in force direction error is much slower during the incompatible surgery compared to the compatible surgery, in agreement with experimental results using these types of perturbations (Fig 2B, [8]).

Another observation from virtual surgery experiments with human participants is that the ability of the baseline muscle synergies to reconstruct the muscle activity decreases more with training under incompatible surgeries than under compatible surgeries (Fig 2D, [8]). This greater decrease in reconstruction quality ($R^2$) indicates that the synergies' structure evolved, with new patterns of muscle activity emerging throughout practice. Our model's simulations reproduce the larger decrease in $R^2$ observed during incompatible surgeries relative to the other two perturbations (Fig 2E), although the magnitude of this decrease is smaller than in the experimental data.

In the following subsections, we present the results of our computational model across three main simulations, summarized in Table 1. Simulation 1 evaluates model performance under the three perturbations, while updating either the control policy, or the synergies, or both. Simulation 2 examines the effect of including or not the updating of the forward model of the musculoskeletal system. Simulation 3 investigates the effect of regularization in the model's behavior. Full simulation details are provided in the Methods section.

## 2.3 Effect of different combinations of the learning rate of the control policy and of the learning rate of the muscle synergies

To further examine the model, in Simulation 1 we investigated the effect of the three perturbations under different combinations of learning rates of the model's adaptive elements. Specifically, we tested combinations where at least one of the control policy matrix $Z$ and muscle synergies matrix $W$ had a nonzero learning rate. The results of Fig 3 show that, during the rotation and compatible surgery perturbations, the force direction error (Fig 3A) and force magnitude error (Fig 3B) decrease more rapidly when both $Z$ and $W$ are updated (*right column*), compared to when only $Z$ (*left column*) or only $W$ (*middle column*) is updated. During the incompatible surgery the error does not decrease when only $Z$ is updated, while a modest decrease is observed when only $W$ is updated, and a faster decrease occurs when both are updated. Because incompatible virtual surgeries perturb the forces generated by the muscles such that the baseline muscle synergies do not span the full force space, updating the synergies' structure becomes necessary to reach all force directions and to reduce the force error. Statistical analyses on the force direction error (Table A in S3 Text) and force magnitude error (Table B in S3 Text) reveal no significant differences between the rotation and the compatible surgery perturbations, but significantly larger errors for the incompatible surgery compared to the other two perturbations, under all three combinations of model parameters and learning rates.

In these simulations, we also examined the ability of the baseline muscle synergies to reconstruct the muscle activity generated by the model over time. Regarding the reconstruction quality ($R^2$) of the muscle activity patterns (Fig 3C), there

**Table 1. Description of main simulations performed with model.**

| Simulation # | Goal of simulation | Parameters investigated |
|---|---|---|
| 1 | Effect of update of the control policy and the muscle synergies | Control policy learning rate 0.05; no update of synergies |
| | | No update of control policy; synergies learning rate 0.05 |
| | | Control policy and synergies learning rate 0.05 |
| 2 | Effect of the update of the forward model | No update of forward model |
| | | Forward model updated with learning rate 0.25 |
| | | Ideal forward model used |
| 3 | Effect of the regularization in the control policy and muscle synergies | No regularization of control policy and muscle synergies |
| | | Regularization weight of control policy and synergies set to 1% of each component's learning rates |

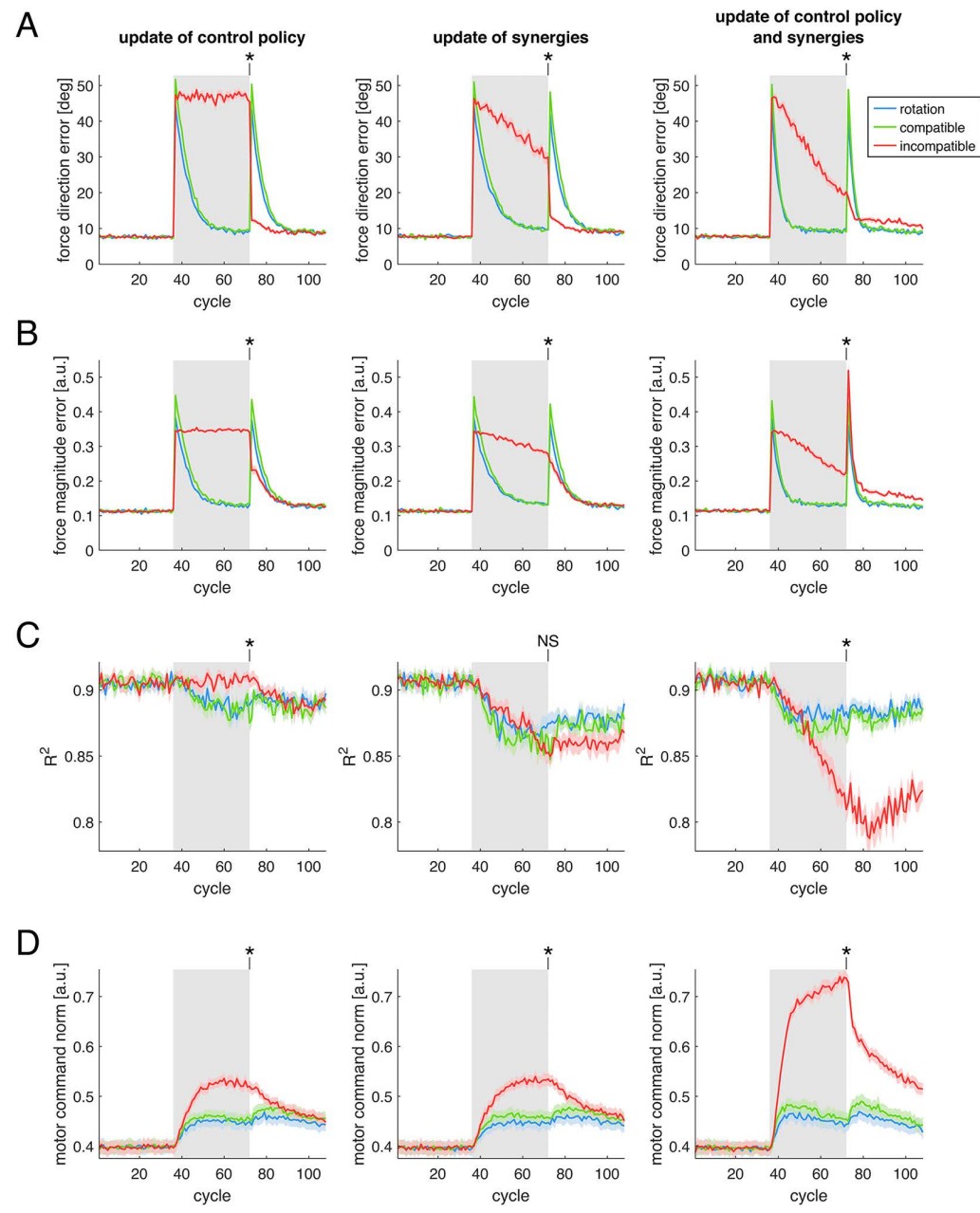

**Fig 3. Effect of different combinations of learning rates.** Different combinations of learning rates of the control policy matrix **Z** and muscle synergies matrix **W** are shown in different *columns*. Each panel *(rows)* shows a different performance metric. **(A)** Force direction error. **(B)** Force magnitude error. **(C)** Reconstruction quality ($R^2$) of the muscle activity using the original synergies. **(D)** Norm of the motor commands. Colored *lines* correspond to the three different types of perturbation simulated. Solid lines and shaded regions indicate average and standard error of each metric across 16 different model initializations. Gray background rectangles indicate cycles during which each perturbation was applied. Asterisks indicate results from statistical tests where a significant difference was found between at least two of the perturbations, at the final cycle of the perturbation (NS: not significant).

is a similar decrease for both the rotation and the compatible surgery perturbations, under all three combinations of learning rates (in each column). Statistical analyses (Table C in S3 Text) confirm that there is no significant difference in the reconstruction quality $R^2$, at the final cycle of perturbation, between the rotation and the compatible surgery perturbations.

Although a decrease in the reconstruction quality $R^2$ would indicate a change of the muscle synergies relative to baseline, such a change is not possible in simulations in which muscle synergies are not allowed to update (Fig 3C, *left column*). As we show in the additional simulation in Section C in S2 Text, the decrease in the $R^2$ observed here, when updating only the control policy (Fig 3C, *left column*), is due to the increase in the norm of the motor commands (Fig 3D): motor commands with a larger norm increase the signal-dependent noise added, which results in a decrease of the signal-to-noise ratio of the underlying muscle synergies, and thus in a decrease in the $R^2$ of the muscle activity reconstruction by the synergies, even though the synergies themselves are not changed.

During the incompatible surgery, the reconstruction quality $R^2$ does not decrease when only the control policy is updated (Fig 3C, *left column*), however it does decrease when the muscle synergies are updated (Fig 3C, *middle and right columns*). This decrease is larger when both control policy and synergies are updated, suggesting a larger change in the synergies compared to the initial ones. When only the control policy is updated (Fig 3C, *left column*), there is a difference in the $R^2$ (Table C in S3 Text) between the incompatible and the compatible surgeries ($p = 2.9 \times 10^{-3}$), but not between the incompatible surgery and the rotation ($p = 0.35$). When only the synergies are updated (Fig 3C, *middle column*), no statistically significant differences are observed between the incompatible surgery and the two other perturbations. Finally, when both control policy and synergies are updated (Fig 3C, *right column*), $R^2$ differs between the incompatible surgery and the two other perturbations, indicating a larger decrease in the reconstruction quality for the incompatible surgery under this parameter combination. A larger decrease in muscle activity reconstruction quality during incompatible surgeries than during compatible surgeries has been observed experimentally [8], although to a greater extent than observed in our simulations.

Regarding the washout period, after removal of the perturbations, we observe a large aftereffect in the force direction error for both the rotation and the compatible surgery, and a comparatively smaller aftereffect for the incompatible surgery. Aftereffects are considered a hallmark of implicit adaptation, having been reported in a variety of motor tasks [16], and a larger aftereffect for compatible surgeries than for incompatible surgeries has been observed experimentally [8]. Although a large aftereffect is present in the force magnitude error for the incompatible surgery, there are no experimental data with which to directly compare this result: in experiments involving virtual surgeries, participants were allowed to make online corrections to their cursor movements [8,36], making it difficult to isolate the feedforward component of the movement and, consequently, to quantify a feedforward force error. We also observe that neither force direction nor force magnitude errors returned to baseline levels after an incompatible surgery when both control policy and synergies were updated, whereas both errors return to baseline when only the synergies are updated. This difference can be attributed to a larger change in the synergies and to a larger muscle activity norm in the former condition. Following larger changes in the synergy structure, their increased recruitment is required to produce the same level of force upon return to baseline conditions, which in turn amplifies the signal-dependent motor noise. As a result, force errors remain elevated and do not return to baseline by the end of the simulation.

After incompatible surgeries, $R^2$ remained at a lower level during washout than after compatible surgeries. Although this effect was not observed in [8], an experiment in which participants first performed a task under an incompatible surgery and subsequently under a compatible surgery reported a persistent decrease in the $R^2$ during the second perturbation [10], consistent with our simulation results. This behavior is replicated in additional simulations in which the model is trained on sequences of incompatible-compatible surgeries, and vice-versa (Section B in S2 Text).

We also examined the norm of the muscle activity generated by the model during training under the different perturbations (Fig 3D). When both the control policy and muscle synergies are updated (*right column*), we observe a transient increase in the norm of the muscle activity during both the visuomotor rotation and the compatible surgery, occurring largely when the force magnitude error is decreasing, followed by a slow decrease. These transient increase and subsequent slow decrease reappear during the washout. In contrast, during the incompatible surgeries, the increase in muscle activity persists throughout the perturbation and decreases only during washout. Statistical analyses (Table D in S3 Text)

 

of the last perturbation cycle reveal significant differences in the norm of the muscle activity for the incompatible surgery compared with the other two perturbations, but no significant differences in the norm of the muscle activity between the rotation and the compatible surgery perturbations, across all combinations of model parameters.

While the norm of the muscle activity has not been examined in previously published experiments involving virtual surgeries [8,10,36], a rapid increase followed by a slight decrease of muscle activity has been observed in arm reaching experiments with velocity-dependent and divergent force fields [37]. We consider these two perturbations to be similar to our rotation and compatible surgery perturbations. In our model, the slow decrease in the muscle activity after the transient increase is primarily driven by the regularization applied to both the control policy and the muscle synergies: this effect becomes more evident in Simulation 3, where we directly compare models trained with and without regularization.

In the condition in which both control policy and synergies are updated, we also evaluated the force prediction error of the forward model. We compared the force predicted by the forward model given a muscle activity generated by the controller and the force executed by the musculoskeletal system, including under perturbation (Fig A in S4 Text). The force prediction error is high at the onset of the perturbations, but decreases as the forward model is updated. Interestingly, this decrease is fastest for the incompatible surgery, although it quickly reaches a plateau at which the decrease slows down. This fast decrease contrasts with the slower decrease in the executed error (Fig 3A and 3B), suggesting that force prediction error alone does not necessarily correlate with learning speed.

## 2.4 Subspaces of muscle activity during training

In the previous section, when both the control policy and the muscle synergies in the model were updated, we showed that 1) the reconstruction quality of the muscle activity by the original synergies ($R^2$) decreased more during the incompatible surgery and 2) the norm of the muscle activity increased more during the incompatible surgery compared to the two other perturbations. These results suggest training under the different perturbations engages different subspaces of the muscle activity space. To elucidate how the muscle activity changes during the perturbations, we evaluated the norm of the muscle activity projected into four different subspaces of the muscle activity space (Fig 4), considering the combination of model parameters when both the control policy and the muscle synergies are updated.

When analyzing the norm of the muscle activity projected in the baseline task space (the subspace of the muscle activity space that affects the force generated in the unperturbed task), we observe an initial increase for all perturbations. However, the norm quickly decreases to near baseline levels for the rotation and the compatible surgery, while remaining elevated for the incompatible surgery (Fig 4A). Statistical analyses of the last perturbation cycle (Table E in S3 Text) reveal a statistically significant difference between the norm of the muscle activity projected in the baseline task space between the incompatible surgery and the two other perturbations, with no significant difference between the rotation and the compatible surgery.

The large increase in the norm of the muscle activity projected in the baseline task space for the incompatible surgery is due to an initially inappropriate forward model. At the beginning of the training, the forward model attempts to correct movement errors caused by the perturbation—which reduces the span of the baseline synergies to a single dimension—by increasing the patterns of muscle activity in the baseline task space [21], consistent with its initialization. This increase in muscle activity decelerates as the forward model adapts to the incompatible surgery, and muscle activity patterns outside the baseline task space are increasingly recruited. The large norm of the muscle activity in the baseline task space immediately before the removal of the incompatible surgery underlies the large after-effect in the force magnitude error described in the previous section (Fig 3). After the removal of the perturbation, the norm of the projection of the muscle activity in the baseline task space transiently increases again for both rotation and compatible surgery, whereas for the incompatible surgery it returns to baseline. This decrease occurs because the environment is the same as the baseline task environment and the required forces are the same, so that the baseline task component of the muscle activity naturally returns to baseline levels for all perturbations.

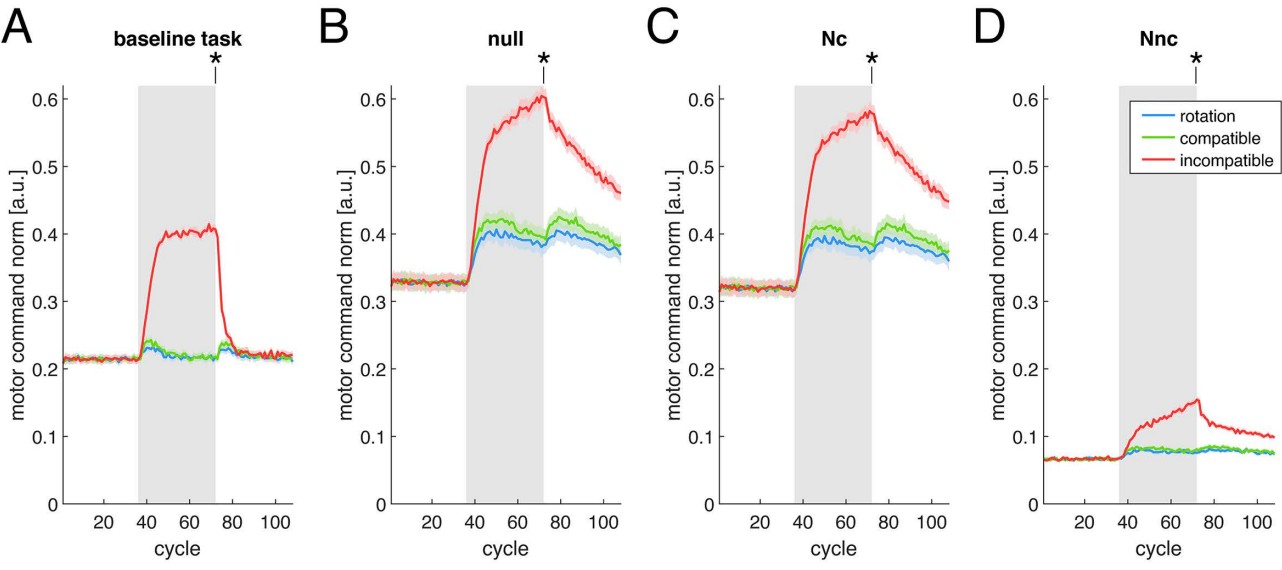

**Fig 4. Norm of muscle activity in different subspaces of the muscle activity space.** Data from Simulation 1, under the model parameter combination with an update of both control policy and muscle synergies. The lines (colored according to the type of perturbation) indicate the norm of the muscle activity projected into different subspaces of the muscle activity space (*panels*), averaged across all targets in a cycle (*gray background rectangles indicate cycles during which each perturbation was applied*). Solid lines and shaded regions are the average and standard error across sixteen different model initializations. **(A)** norm in the baseline task space, the subspace that affects the force in the baseline (not perturbed) task. **(B)** norm in the null space of the baseline task space. **(C)** norm in Nc, the common subspace between null space and baseline synergies. **(D)** norm in Nnc, the subspace of null space not spanned by baseline synergies. Asterisks indicate results from statistical tests where a significant difference was found between at least two of the perturbations, at the final cycle of the perturbation.

Noticeably, when analyzing the norm of the muscle activity in the null space (Fig 4B) and in its $N_c$ (intersection between null space and baseline synergies) and $N_{nc}$ (subspace of null space not spanned by baseline synergies) subspaces (Fig 4C and 4D, respectively), we observe that most of the activity lies in the $N_c$ space, while the component of the muscle activity in the $N_{nc}$ space is much smaller. In our modular architecture, muscle activity is generated by a combination of synergies, so at baseline most of the muscle activity in the null space is explained by synergy combinations (i.e., the $N_c$ space), whereas activity in the null space not spanned by the baseline muscle synergies (i.e., the $N_{nc}$ space) is mostly due to motor noise. We see an increase in the muscle activity in the $N_{nc}$ space at the onset of the incompatible surgery, while its increase during the rotation and the compatible surgery is negligible. This is confirmed by a statistically significant difference in the norm of the muscle activity in the $N_{nc}$ space between the incompatible surgery and the two other perturbations (Table E in S3 Text). Unlike the norm of the muscle activity in the baseline task space, which initially rises at the onset of the incompatible surgery and then plateaus, the norm in the $N_{nc}$ space continues to increase throughout the entire training period (Fig B in S4 Text). After the removal of the perturbation, the norm of the muscle activity in the $N_{nc}$ space slowly decreases for the incompatible surgery, but does not fully return to the baseline level at the end of the washout phase.

In summary, analysis of muscle activity norms across different subspaces shows that, during the perturbation, activity in the null space not spanned by the baseline synergies (the $N_{nc}$ space) increases more for the incompatible surgery compared with the two other perturbations. These learning-related changes in the patterns of muscle activity persist for some time even after the removal of the perturbation, and are consistent with our earlier finding that the reconstruction quality of the muscle activity decreases more for the incompatible surgery than for the compatible surgery.

## 2.5 Changes in the structure of muscle synergies during training

In the previous section, we described differences in the evolution of muscle activity during training under the three different perturbations, suggesting substantial reorganization of the muscle synergies during incompatible surgeries. Our computational model enables direct analysis of these structural changes by examining the muscle synergy matrices $\boldsymbol{W}$ throughout the simulations. To quantify changes in the synergy structure in Simulation 1—where all model components had nonzero learning rates—we used two additional metrics: the first captures the span of the synergy forces within different task environments while the second assesses the alignment of the synergies with specific directions in muscle activity space.

The first metric, the *area of the convex hull of the synergy forces*, was computed in both the baseline task space and in the incompatible surgery task space (Fig 5A) to assess how well the synergies span the force space. In the incompatible task space, the area of the baseline synergies is zero because the incompatible surgery is designed so that the baseline synergies do not span the entire force space, resulting in a convex hull corresponding to a straight line, with zero area. In experiments with human participants, the area in the baseline task space decreases, while the area in the incompatible task space increases during training with an incompatible surgery [36], indicating that synergies are restructured toward the dimensions relevant to the incompatible task space.

Our simulations reproduce this pattern (Fig 5A). During the incompatible surgery, the area in the baseline task space decreases and becomes smaller than during the rotation perturbation, though it is not significantly different from the compatible surgery (Table F in S3 Text). In contrast, the rotation and compatible surgery show an initial decrease followed by an increase, suggesting a transient change in the span of the synergies in the baseline task space. In the incompatible task space, the convex hull area of the synergy forces increases substantially and continuously under the incompatible surgery, whereas in the other two perturbations this increase is modest and stabilizes at a significantly smaller value (Table F in S3 Text). Together, these results indicate a progressive reorganization of the muscle synergies during training under the incompatible surgery, enabling them to span a larger area of the incompatible task space.

The second metric, the *principal angle* between the synergies and the vectors $\{\boldsymbol{w}, \boldsymbol{w}', \boldsymbol{n}\}$, quantifies how the synergy structure aligns with the directions used to define the virtual surgeries (Fig 5B). These three vectors are used to define the compatible and incompatible virtual surgeries. In the compatible surgery, the vector $\boldsymbol{w}$ in the synergy subspace orthogonal to the null space (the $\boldsymbol{W}_{nc}$ space) is rotated towards the vector $\boldsymbol{w}'$ within the same subspace. In the incompatible surgery, the vector $\boldsymbol{w}$ is rotated towards the vector $\boldsymbol{n}$ in the subspace of the null space not spanned by the baseline synergies (the $\boldsymbol{N}_{nc}$ subspace). At the beginning of training, the principal angles with $\boldsymbol{w}$ and $\boldsymbol{w}'$ are zero, as these vectors lie within the baseline synergies subspace, and an increase in these angles indicates a deviation from the original synergy structure. In contrast, the principal angle with $\boldsymbol{n}$ is initially larger than zero, since $\boldsymbol{n}$ lies in the $\boldsymbol{N}_{nc}$ subspace not spanned by the baseline synergies, and a decrease in this angle indicates a reorganization of the synergies toward the $\boldsymbol{N}_{nc}$ space.

Our simulations show increases in the principal angles between the synergies and both vectors $\boldsymbol{w}$ and $\boldsymbol{w}'$ across the three perturbations (Fig 5B, *left and middle panels*). However, the increase in the principal angle with $\boldsymbol{w}$ (the vector from which the muscle activity space is rotated during the virtual surgeries) is significantly larger during the incompatible surgery than during the other two perturbations (Table G in S3 Text). Interestingly, the increase in the principal angle with $\boldsymbol{w}'$ (the vector toward which the muscle activity space is rotated during the compatible surgery) is significantly smaller during the incompatible surgery than during the two other perturbations (Table G in S3 Text). These results indicate that synergy structure changes occur during all perturbations, but during the incompatible surgery synergies deviate more strongly from the direction not relevant to the incompatible surgery task ($\boldsymbol{w}$) and less from directions unaffected by it ($\boldsymbol{w}'$).

Regarding the principal angle between the synergies and the vector $\boldsymbol{n}$ (Fig 5B, *right panel*), we observe a decrease across all perturbations, with a larger reduction for the incompatible surgery compared to the rotation, though not significantly different from the compatible surgery (Table G in S3 Text). For both the rotation and compatible surgery, the decrease in the principal angle tends to plateau during training (as seen with the other two vectors), whereas for the

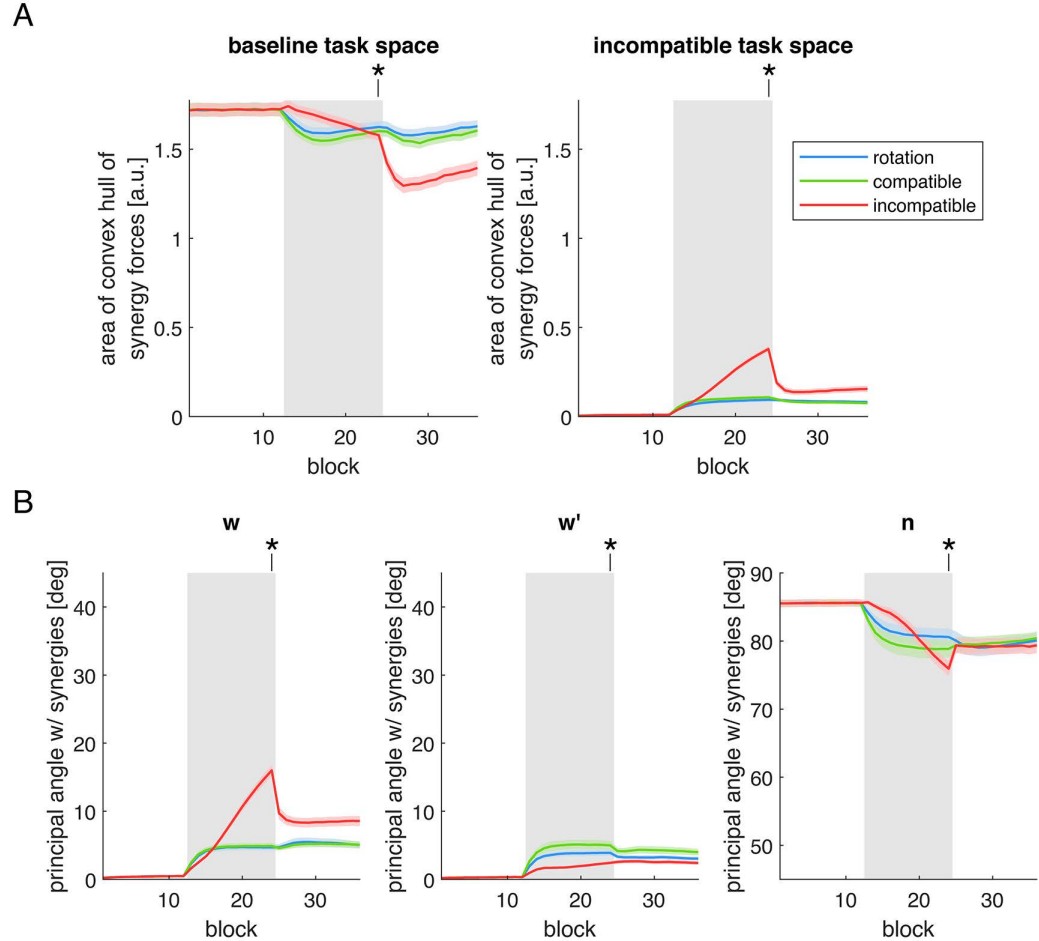

**Fig 5. Metrics related to the change in the structure of muscle synergies.** Data from Simulation 1, under the model parameter combination where both control policy and muscle synergies were updated. Colored lines indicate the metrics calculated from the synergies saved at the end of each block (three cycles), with gray background rectangles marking the perturbation periods. Solid lines and shaded regions denote the average and standard error across sixteen model initializations. **(A)** Area of the convex hull of the synergy forces, calculated in the baseline task space (*left*) and in the incompatible task space (*right*). **(B)** Principal angles between the synergies and three vectors in the muscle activity space used to construct the virtual surgeries: (*left*) **w**, a vector in the synergy subspace not in the null space, from which the muscle activity space is rotated in the virtual surgeries; (*middle*) **w′**, another vector in the synergy subspace not in the null space, toward which the muscle activity space is rotated in the compatible surgery; (*right*) and **n**, a vector in the null space not spanned by the baseline synergies, toward which the muscle activity space is rotated in the incompatible surgery. Asterisks indicate statistically significant results (at 5% significance level) between at least two of the perturbations, at the final block of perturbation.

incompatible surgery the decrease continues steadily throughout the perturbation. These findings contrast with the results of the previous section, which showed a greater increase in the muscle activity norm in the $N_{nc}$ space during incompatible surgeries. This suggests that, despite a similar alignment with vector **n**, the recruitment of these better aligned synergies is stronger during the incompatible surgery, resulting in a higher muscle activity norm.

During the washout period, the area of the convex hull of the synergy forces in the baseline task space (Fig 5A, *left panel*) remains lower than during the baseline period, for all three perturbations, and is lower after the incompatible surgery than after the two other perturbations. Likewise, the area in the incompatible task space (Fig 5A, *right panel*) does not return to baseline, remaining larger after the incompatible surgery than after the two other perturbations. A similar trend can be observed for the principal angles between the synergies and the vectors **w**, **w′**, and **n** (Fig 5B), which also do

not return to baseline levels after any of the perturbations. These results indicate that changes in the structure of the synergies, which are more pronounced during the exposure to the incompatible surgery, persist even after the perturbation is removed. This accounts for the persistent decrease in the reconstruction quality ($R^2$) of muscle activity using the original muscle synergies and for the persistent increase in the norm of the muscle activity in the $\boldsymbol{N}_{nc}$ space after the incompatible surgery.

In summary, our analyses show that the structure of muscle synergies evolves under all three perturbations. While changes are similar for the visuomotor rotation and the compatible surgery, the incompatible surgery induces more pronounced structural modifications: the synergies deviate further from the direction irrelevant to the task and expand their forces within the incompatible task space. This larger, persistent reorganization of synergy structure, together with changes in the synergy recruitment, explains the greater reduction in the reconstruction quality ($R^2$) of muscle activity using the original muscle synergies observed in our simulations of incompatible surgeries.

## 2.6 Examining different learning rates of the forward model

In the previous simulation, the forward model of the musculoskeletal system was updated as the models encountered the perturbations. In contrast, previous computational models of the trial-by-trial update of motor activity which included a modular architecture assumed that the forward model, responsible for the error correction, was either known *a priori*, or instantly fully learned as soon as the perturbation was introduced [22,23], an assumption that has been criticized as implausible [32]. In Simulation 2, we investigated three possibilities for the learning of the forward model: no update of the forward model (learning rate of zero), a slowly updated forward model (non-zero learning rate, the same one used in Simulation 1), or the ideal forward model for each perturbation. This allows us to compare the predictions of our model with those of models assuming an ideal forward model, as well as with available experimental data.

The results of Simulation 2 (Fig 6), show no apparent difference in any of the metrics during the rotation and the compatible surgery across all three learning rates of the forward model (*each column*). Statistical analyses on the force direction error (Fig 6A and Table H in S3 Text), force magnitude error (Fig 6B and Table I in S3 Text), reconstruction quality of the muscle activity using the original muscle synergies ($R^2$) (Fig 6C and Table J in S3 Text), and norm of the muscle activity (Fig 6D and Table K in S3 Text) at the final cycle of perturbation confirm that there is no significant difference across any of the parameters for the forward model for either perturbations.

For the incompatible surgery, in contrast, clear differences emerge across the three learning rates for all four metrics. In comparison to when the forward model is updated slowly (*middle column*), both force direction error (Fig 6A) and force magnitude error (Fig 6B) are substantially higher when the forward model is not updated (*left column*) (even higher than at the onset of the perturbation), and markedly lower when the ideal forward model is used (*right column*). In the latter condition, the decrease in the error during the incompatible surgery becomes comparable to that observed for the two other perturbations, with the exception of the washout phase: a transient increase in error is present for the rotation and the compatible surgery but absent for the incompatible surgery. In terms of reconstruction quality ($R^2$, Fig 6C), there is no significant difference in the decrease between the slowly updated (*middle column*) and not-updated (*left column*) forward model conditions (Table J in S3 Text). However, there is a significantly larger decrease when the ideal forward model is used (*right column*), compared to the two other conditions. Similarly, the motor command norm (Fig 6D) reaches a significantly higher level when the forward model is not updated (*left column*), and a significantly lower level when the ideal forward model is used (*right column*), relative to the slowly updated condition (*middle column*, Table K in S3 Text).

When the forward model is not updated, the incompatible surgery leads to a large increase in the norm of the muscle activity generated by the model (Fig 6D, *left column*), while the error in the force generated by this muscle activity does not decrease and instead increases relative to perturbation onset (Fig 6A and 6B). Because updates to the muscle activity are dictated by the forward model, and incompatible surgeries require novel patterns of muscle activity, changes in the

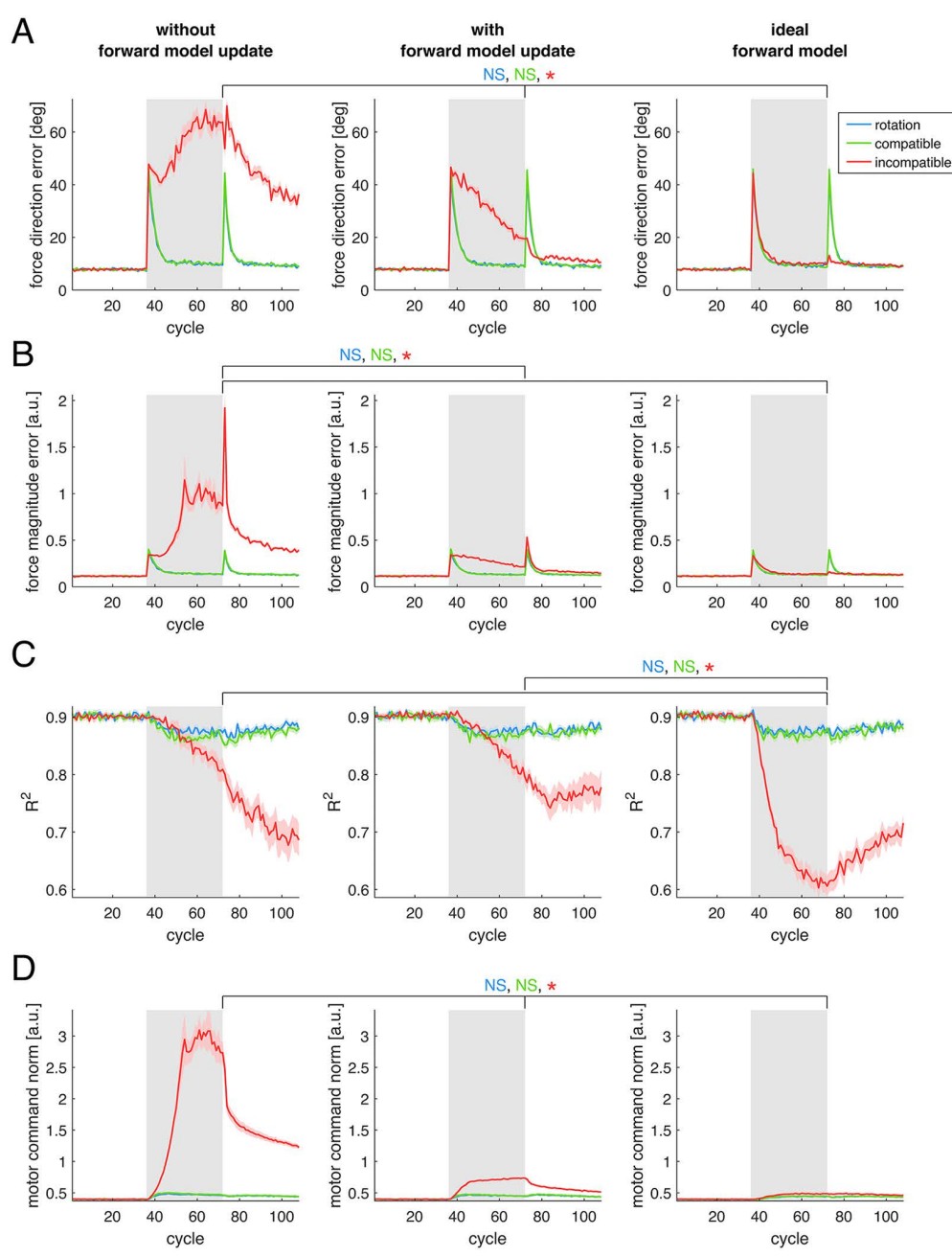

**Fig 6. Effect of different learning rates of the forward model.** Three different learning rates of the forward model are presented in different *columns*. Colored lines correspond to the three different perturbations simulated. Solid lines and shaded regions are the average and standard error of each metric (*rows/panels*) across sixteen different model initializations. **(A)** Force direction error. **(B)** Force magnitude error. **(C)** Reconstruction quality ($R^2$) of the muscle activity using the original synergies. **(D)** Norm of the motor commands. Gray background rectangles indicate cycles during which each perturbation was applied. Asterisks (NS: not significant) indicate results from statistical tests where a significant difference was found between at least two of the conditions, with the text colors indicating the perturbation tested, at the final cycle of the perturbation.

muscle activity generated by the baseline forward model are inappropriate for the perturbed environment. As a result, the increased muscle activity norm fails to reduce the force error.

In contrast, with an ideal forward model (*right column*), the force error during the incompatible surgery decreases as fast as during the rotation and the compatible virtual surgery, accompanied by a large change in the structure of the muscle activity, as reflected by the pronounced decrease in $R^2$. Analysis of the norm of the projection of the muscle activity in different subspaces when using the ideal forward model (Fig C in S4 Text) reveals a large increase in the use of the $N_{nc}$ space during the incompatible surgery, and no increase in the norm on the baseline (unperturbed) task space. This contrasts with the condition in which the forward model is slowly updated, where the norm of the muscle activity in the baseline task space increases markedly (Fig 4A), and the decrease in $R^2$ is smaller (Fig 6C, *middle column*). In addition, the force magnitude error (Fig 6B) at the onset of the washout (after the removal of the perturbation) is much smaller when an ideal forward model is used (*right column*), reflecting the smaller increase of the norm of the muscle activity in the baseline task space (Fig C in S4 Text) in this condition. This phenomenon is similar to the encoding of different motor tasks in orthogonal neural dimensions, which reduces interference between motor memories [38]. These results further indicate that, during an incompatible surgery, changes in the baseline (unperturbed) task component of the muscle activity arise primarily from an incorrect forward model. A correct forward model instead leads to 1) larger changes in the muscle activity, specifically in the $N_{nc}$ space; 2) a faster decrease of the force error; and 3) a greater separability between familiar and novel motor tasks, resulting in a smaller washout force magnitude error.

## 2.7 Effect of regularization

In the preceding simulations of our model, whenever a model component (either the control policy matrix **Z** or the muscle synergies matrix **W**) was updated, a nonzero regularization was applied, pulling matrices entries towards zero. In Simulation 3, we compared two combinations of parameters of the model, with and without regularization, to characterize the effect of regularization. Overall, regularization produced small but statistically significant effects on force direction error (Fig 7A), force magnitude error (Fig 7B), and reconstruction quality ($R^2$, Fig 7C), across all perturbations. Statistical analyses at the last cycle of perturbation (Table L in S3 Text) revealed a significant average reduction of 1.67° in the force direction error and of 0.024 (arbitrary units) in the force magnitude error, along with a significant increase of 0.019 in $R^2$ caused by the regularization. Regularization also had a significant negative effect on the motor command norm (Fig 7D and Table M in S3 Text), with a significant interaction with the perturbation type. This effect was stronger for the incompatible surgery ($F(1,90) = 410.00$) than for the compatible surgery ($F(1,90) = 112.00$) and the rotation ($F(1,90) = 103.54$). The stronger effect of the regularization in the incompatible surgery may be due to a smaller increase in the norm of the motor commands during the other two perturbations.

These results suggest that, although regularization significantly affects the force magnitude error, its largest effect is the reduction of the norm of the muscle activity generated by the model. As there is evidence for reduction of effort at muscular level [37,39] and at synergy recruitment level [40], as well as reduction of metabolic cost during prolonged practice [41], the inclusion of a regularization term in our computational model is well justified.

## 3 Discussion

Although the redundancy of the human body affords the central nervous system infinitely many possibilities for coordinating multiple joints during movement execution [1], strong regularities are observed in the recruitment of muscles [3]. These regularities can be explained by the existence of a small number of control modules that can be flexibly combined during force generation, thereby simplifying coordination across the many degrees-of-freedom of the musculoskeletal system [2]. A modular architecture for the generation of muscle activity patterns also accounts for differences in the learning speed under different perturbations, depending on whether the perturbation can be compensated for using the original control modules [8]. However, computational models of error-based learning of novel muscle activity patterns either did

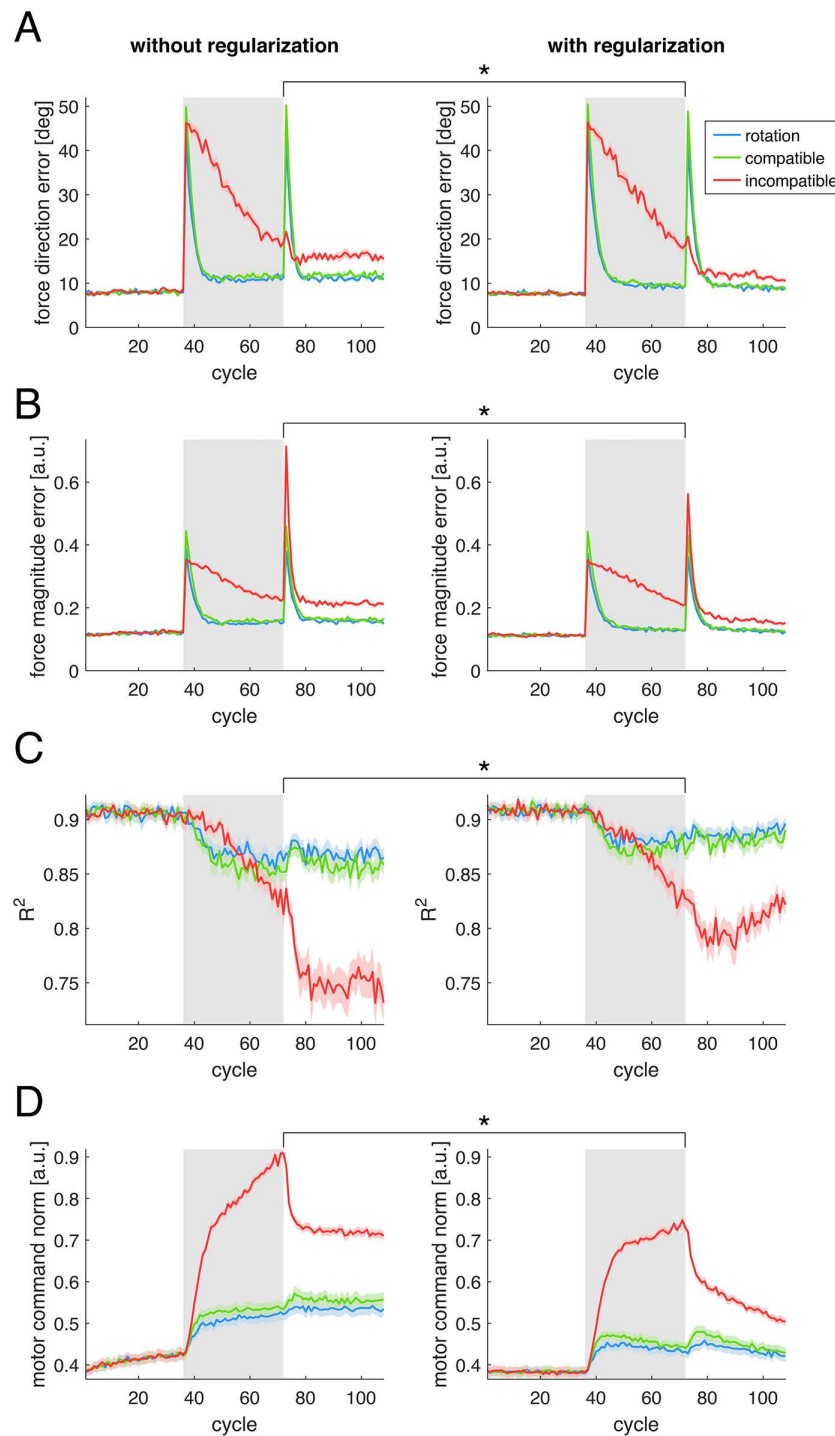

**Fig 7. Effect of regularization.** Simulations performed with or without regularization are shown in different *columns*. Colored lines correspond to the three different types of perturbation simulated. Lines and shaded regions are the average and standard error of each metric (on each *row/panel*) across sixteen different model initializations. **(A)** Force direction error. **(B)** Force magnitude error. **(C)** Reconstruction quality ($R^2$) of the muscle activity using the original synergies. **(D)** Norm of the motor commands. Gray background rectangles indicate cycles during which each perturbation was applied. Asterisks indicate results from statistical tests where a significant effect of the regularization was found, examined at the final cycle of the perturbation.

not account for this modularity [20,21], or incorporated it under the assumptions of ideal error correction or immutable modules [22,23].

We proposed a modular architecture for the trial-by-trial generation and learning of redundant muscle activity patterns during isometric force-reaching tasks. In our computational model, explicitly defined modules, representing spatial muscle synergies, are recruited by a control policy and are combined for the generation of muscle activity, which serves as the input to the musculoskeletal system and results in the generation of force. Both the muscle synergies and the control policy are updated, at different learning rates, by backpropagation of the force error through a forward model of the musculoskeletal system, which is not assumed to be known *a priori* and is instead updated using force prediction error. Our model qualitatively reproduces the main findings of experiments involving virtual surgeries that alter the forces generated by the muscle activity [8,10], supporting the hypothesis that the generation and learning of muscle activity patterns in humans rely on a modular architecture. Nevertheless, discrepancies between our simulations and available experimental data were also observed.

### 3.1 Error-based learning of new muscle activity patterns requires updating internal model for error correction

In our computational model, a forward model of the musculoskeletal system is responsible for the correction of movement errors, and this model is updated through force prediction error. During goal-directed movement, the goal of the movement is typically specified in terms of distal variables, such as a specific posture of the body in space, or the position and orientation of a tool that is being manipulated. However, the CNS cannot control these distal variables directly, and instead relies on the control of proximal variables, such as muscle activity, to achieve movement goals. When a movement error occurs—defined as a discrepancy between the desired and executed movement— a "distal teacher" may transform this distal error into an appropriate change in proximal variables, thereby enabling updates to the control policy [17]. It has been argued that the knowledge of this "teacher" responsible for error correction cannot be innate and must instead be learned through experience [32]. Accordingly, in novel sensorimotor tasks that require new patterns of muscle activity, such as incompatible virtual surgeries, an internal model for error correction must be acquired for learning to occur [21]. Consistent with this view, our simulations show that incompatible virtual surgeries cannot be learned when the forward model is not updated. These findings support the inclusion of a gradually learned forward model of the musculoskeletal system as a key component of our model.

We have shown that, in our model, the updates of both the control policy and muscle synergies depend on the structure of the forward model. The closer the forward model is to the ideal (i.e., true) forward model, the faster the error decreases, and the more substantial the reorganization of muscle activity during incompatible surgeries. Learning an appropriate forward model under incompatible surgeries requires experience with motor commands that lie in the portion of the null space not spanned by the original muscle synergies (the $N_{nc}$ space). However, generating such examples is constrained by the structure of the existing synergies. Previous work has proposed that motor exploration facilitates forward model learning [20]. Consistent with this idea, comparing Simulation 1 (Fig 4), in which motor noise was included and the forward model was updated, with the simulation in Section C of S2 Text, in which no motor noise was added, reveals a faster decrease in error during incompatible surgeries when motor noise is present. These findings support a role for motor exploration in enabling forward-model learning and the subsequent reorganization of muscle activity through error-based learning.

While motor noise at the muscle output level enhances error correction in our model, introducing noise at other levels of the model hierarchy may not yield similar benefits. Noise at the synergy recruitment level could constrain exploration to the baseline synergies, thereby limiting learning during incompatible surgeries. Noises in the internal components of our model (the control policy and the synergy matrices), while important for reward-based algorithms such as weight and node perturbation [42], would simply accumulate at the motor output level in our model, producing effects similar to the currently implemented signal-dependent motor noise; this justifies our simplified approach. Noise that the CNS is not aware of

PLOS Computational Biology

(i.e., execution, rather than exploratory noise), such as in the feedback errors driving learning, could be detrimental and may require adjustment of the learning rates [43], a mechanism not implemented in our model. Further work is needed to examine the influence of different sources of noise within the model, potentially combined with adaptive regulation of learning rates according to noise variability [44].

### 3.2 Contextualizing our computational model within existing models and experimental data

The behavior of our model during adaptation to compatible virtual surgeries (where the baseline synergies still span the entire force space) resembles adaptation to a visuomotor rotation. Although visuomotor rotations have not, to our knowledge, been studied in the context of myoelectric control, experiments with human participants have shown that adaptation to force perturbations (which, like visuomotor rotations, do not act directly at the muscle level) is easier than adaptation to compatible surgeries [8]. This difference may arise because experimental virtual surgeries are constructed using estimates of each participant's synergies, allowing residual muscle activity not captured by those synergies to contribute significantly to task performance [45]. In contrast, our simulations defined virtual surgeries using the "ground-truth" synergies. Experimentally, adaptation to compatible surgeries produces only a small decrease in reconstruction quality of the muscle activity using the original synergies ($R^2$), similar to adaptation to force perturbations [8], a result that is replicated in our simulations when comparing compatible surgeries with visuomotor rotations (Fig 3C).

Computational models of trial-by-trial learning that included modularity in the generation of muscle activity have been investigated before. Hagio & Kouzaki (2018) showed that modularity may accelerate learning by reducing the bias of the distribution of the mechanical contributions of neurons in the controller neural network. A similar argument has been made by Barradas et al. (2023), where the speed of learning under a perturbed environment was related to the uniformity of the shape of the cost function, expressed as the ratio between the smallest and the largest eigenvalue of the Hessian matrix of the cost function around its minimum. They reported that, in a non-modular architecture using the distal learning framework, learning after an incompatible surgery could be slightly faster than learning after a compatible surgery, due to differences in the uniformity of the cost functions across the two virtual surgeries. In contrast, in a modular architecture, learning after incompatible surgeries was slower than after compatible surgeries. There are key differences between our simulations and those of Barradas et al. (2023) that must be considered: 1) we do not assume that the forward model responsible for the error correction is known at the beginning of the perturbation; it is learned by minimizing the force prediction error; 2) we assume that the structure of the spatial muscle synergies can be updated by the learning algorithm, whereas they assumed fixed synergies; and 3) we used "ground truth" synergies to define the virtual surgeries, while they used estimated synergies from the muscle activity generated by the model during baseline. Using an estimate of the synergies to define incompatible surgeries may cause them being not truly "incompatible" (in the sense that they may still allow the existing synergies to span the entire force space), although this approach has the merit of following more closely the experimental approach, in which the ground-truth synergies are unknown. Additional simulations using a large number of fixed synergies and using estimated synergies to define the incompatible surgery showed only a small decrease in $R^2$ but a decrease in the error (see Section A in S2 Text), suggesting that allowing synergies to change over time provides a better fit to experimental observations.

The regularization implemented in our model, applied to both the control policy and synergy matrices, significantly reduced the norm of the motor commands, corresponding to the reduction of muscular [37,39] and metabolic [41] effort observed in human motor tasks. Our implementation of the regularization is similar to the weight decay algorithm widely used in artificial neural networks to improve generalization [46]. Weight decay has also been investigated in computational models of sensorimotor transformations [47], where it has been shown that, over tens of thousands of trials of training (with regularization weights comparable with those used in our simulations), effort can be reduced to near-optimal levels. In contrast, our simulations involved a number of trials similar to a typical single-session human experiment—on the order of hundreds of trials—and were therefore likely too short to achieve full minimization of effort. In addition, experiments with

humans performing three-dimensional isometric force tasks suggest that it is not the individual muscle activity, but rather the recruitment of muscle synergies that is minimized, albeit sub-optimally [40]. Further experiments are needed to clarify how effort is minimized at both muscular and synergistic levels during force generation and learning.

### 3.3 Possible neural substrates of motor adaptation with synergies

The components of our model may be linked to specific neural substrates. Evidence from spinal and cortical microstimulation [4,48] and single-unit recordings [49] studies in animals suggest that spatial muscle synergies may be encoded in the cortico-spinal connectivity and in spinal or brainstem networks. In humans, the preservation of muscle synergies following unilateral ischemic lesions in frontal motor cortical areas suggests that synergy recruitment may be encoded in higher motor areas [50]. The cerebellum has been implicated in the temporal organization of synergies [51], and is widely associated with prediction errors and forward models [12]. The integrity of the cerebello-thalamo-cortical pathway is related to the capacity to adapt to movement errors [52], but not with the execution of already learned movements [53], consistent with its role as an error-correcting forward model in our architecture. Future experiments and simulations may further elucidate the neural correlates of the components of our model and leverage it as a tool to investigate recovery processes following neural injury.

While the lack of the generation of entire movement trajectories (as discussed in the Limitations section) precludes directly fitting our model to the data from virtual surgery experiments—where participants could make use of online feedback corrections—our model may nonetheless be useful for understanding inter-individual differences in motor learning. Specifically, we predict that participants who exhibit a faster reduction in error during incompatible surgeries would be characterized, in terms of model parameters, by larger learning rates of either the forward model or the synergies. Because the goal of the present work was to characterize how the individual components of the model shape its learning behavior, we defer such analyses to future studies.

In summary, our computational model accounts for differences in adaptation to different types of virtual surgeries—at the task, muscle, and synergy levels—through the learning of three key components of a modular control architecture: the recruitment of muscle synergies, the structure of the synergies, and an acquired forward model of the task and motor system. By systematically analyzing how each component shapes the model's behavior, we provide a framework for linking patient-specific impairments to deficits in the learning or updating of specific elements in our model and to specific neural substrates. In the future, this model may be used to predict inter-individual differences in motor learning in both patients and healthy participants, and to guide the development of targeted interventions aimed at enhancing recovery from specific impairments.

### 3.4 Limitations of our model

One of the limitations of our model is the use of a fixed number of muscle synergies. Changes in the number of synergies involved in locomotion, resulting from synergy fractionation and merging, have been reported during development and training [54]. Fractionation and merging of synergies have been observed after stroke, in the affected arm—relative to the unaffected arm—[55], suggesting that cortical damage can alter both the number and the structure of muscle synergies underlying upper limb movements. Although assuming a fixed number of synergies may be appropriate to study motor learning over short time scales, such as single-session experiments, extending the model to allow for a variable number of muscle synergies could provide a more comprehensive account of changes in motor coordination across development, long-term skill acquisition, and motor rehabilitation.

While we used fixed learning rates in our simulations, studies of motor adaptation in humans suggest that the CNS adjusts its adaptation rate according to the uncertainty of sensory feedback [43] and the consistency of the task environment [56,57]. These findings have been attributed to a metacognitive process that monitors task performance and regulates the rates of retention and adaptation [44]. Incorporating adaptive learning rates into our model could therefore allow

it to reproduce these behaviors and enable a more detailed analysis of changes in muscle activity and synergy structure across a wider range of simulated motor tasks.

We did not include in our model any cognitive processes, despite evidence that such processes influence motor adaptation [58]. In a recent experiment in which participants had more time to reach the target during the virtual surgeries [36], a larger aftereffect in the direction error of the initial movement was observed for the compatible surgery compared to the incompatible surgery immediately after the removal of the perturbations. Large aftereffects after the compatible surgery, a result which is in line with the predictions of our model regarding the force direction error, are a hallmark for implicit adaptation [59]. In contrast, training under incompatible surgeries was associated with longer reaction times [36], suggesting a greater reliance on explicit strategies. Although our computational model predicts a decrease in the reconstruction quality of the muscle activity using the original synergies ($R^2$), this decrease is smaller than what has been observed experimentally [8,36], and the recovery of $R^2$ during the washout phase is also slower than in the experimental data. These discrepancies may indicate that our model lacks a cognitive component that could be responsible for the regulation of motor exploration [60–62]. However, it remains unclear whether cognitive processes would be capable of recruiting muscle patterns that cannot be explained by existing muscle synergies.

As previously noted, the decrease in the reconstruction quality of the muscle activity using the original synergies ($R^2$) during incompatible virtual surgeries in our model is smaller than the decrease observed in experiments with human participants [8,36]. Beyond the possible influence of cognitively-driven exploration strategies discussed above, several additional factors may account for this discrepancy. One possibility is the recruitment of pre-existing muscle synergies that were not identified during the baseline task, which could explain the larger decrease in the $R^2$ observed experimentally. However, simulations in which we increased the number of synergies did not reproduce such a pronounced decrease (Section A in S2 Text). If such latent synergies do exist, our results suggest that their recruitment may rely on mechanisms not captured by our model—possibly involving cognitive processes—during incompatible surgeries.

Another factor concerns how the $R^2$ was measured in experimental studies: in several cases it was computed over the entire movement, whereas when computed separately for the feedforward phase, the decrease was smaller [36]. This result is more in line with our predictions, as we model only the generation and updating of feedforward movement components. In addition, recent experiments using high-density surface electromyography (HDsEMG) have revealed that modularity may also exist at the level of motor neurons [63]. Such motor neuron synergies would provide the CNS with greater flexibility in muscle recruitment and might account for part of the decrease in $R^2$ during incompatible surgeries. While muscle synergies and motor neuron synergies may capture the same modular organization at different levels—functional and neural implementation, respectively— not all existing modules may be identified from the decomposition of bipolar EMG signals, and the unidentified synergies may affect the reconstruction quality. These aspects should be considered when comparing our simulation results to experimental data, and will be further explored in future works.

Another limitation in our model is the assumption that the forward model has the same structure as the muscle activation-to-force mapping (a $D$ x $M$ matrix). We did not investigate other architectures and how they might affect error correction. For example, if the forward model were implemented using radial basis functions, updates might have more localized effects, potentially limiting the generalization of learning across perturbations. Similarly, if the forward model mapped synergy activation coefficients directly to predicted forces, it would implicitly contain an internal representation of the synergies. In that case changes in synergy structure would require corresponding updates to the forward model itself. Error backpropagation would then be confined to the synergy activations—unless a muscle activity error signal could be extracted from an intermediate layer of the forward model. Investigating how different forward-model architectures affect learning and error correction is an important direction for future work.

Finally, our model can only predict the generation of feedforward motor commands and the trial-by-trial learning processes of the muscle activity, whereas movements executed by the motor system are generally continuous and learning may occur throughout the movement duration [36,64–67]. We plan to extend the model to predict the entire time-course

of the muscle activity and of the generated force, as well as to incorporate online feedback error correction mechanisms. A feedback-driven modular architecture in a novel redundant sensorimotor task, without assuming *a priori* knowledge of the sensorimotor mapping, has been investigated recently [68]. However, that model did not account for possible changes in synergy structure, only in their recruitment, and did not include a feedforward component. Experimental evidence from virtual surgery paradigms further supports the need to model both processes: increasing trial duration leads to faster improvements in online feedback error correction than in initial movement error—reflecting trial-by-trial feedforward learning—during incompatible surgeries [36]. This enhanced feedback-based improvement has been attributed to increased exploration enabled by longer movement durations. These findings suggest a possible dissociation between feedback and feedforward error correction mechanisms, which has also been observed in mirror-reversal experiments [69]. From a modeling perspective, this dissociation may imply different gains for feedforward and feedback error correction, or even distinct learning mechanisms with separate learning rates governing trial-by-trial feedforward adaptation and online feedback corrections. Extending our model to generate the entire movement trajectories would allow us to directly investigate these hypotheses and better capture the dynamics of motor learning observed experimentally.

## 4. Methods

### 4.1 Model simulations

We performed simulations to validate the model described in the Results by assessing whether it can adapt to perturbations such as visuomotor rotations and 'virtual surgeries' and whether it can reproduce key features of motor adaptation observed in experiments with human participants. All simulations and analyses were conducted in MATLAB® (MathWorks Inc., Natick, MA). In the simulations, we used $D = 2$ force dimensions, $M = 10$ muscles, and $N = 5$ muscle synergies, values similar in magnitude to the dimension of the muscle activity and number of synergies reported in human experiments [8]. The muscle activation-to-force matrix $H$ was initialized by assigning each of the $M$ muscles a random $D$-dimensional force vector, ensuring that the forces positively spanned the entire force space [70]. The norm of each vector was randomly selected from the interval [0.5, 1.5] (arbitrary units). We defined 8 target forces $f^*$, uniformly distributed in a circle with radius 0.5 (arbitrary units) centered at the origin of the force space (at rest). The control policy was implemented using $N_\Phi = 121$ Gaussian basis functions $\Phi$ with centers uniformly distributed on a 11-by-11 grid spanning the range [-1, +1] in each force dimension. This basis set was sufficient to represent all force targets used in the task. Each element of the matrix $Z$ of the control policy was initialized by sampling from a uniform distribution over the interval [0,0.05]. For the initialization of the synergy matrix $W$, we first generated $N$ random force vectors positively spanning the entire force space [70]. For each of these force vectors, we used quadratic programming (MATLAB® function *quadprog*) to compute the minimum-norm, non-negative muscle activation pattern that produced the desired force under the mapping defined by $H$. These muscle activity vectors, each representing a muscle synergy, were concatenated to form the matrix $W$.

Our model was trained under three different types of perturbation: visuomotor rotations, compatible virtual surgeries, and incompatible virtual surgeries (Fig 2A). Visuomotor rotations applied a 45° counter-clockwise rotation to the force $f$ executed by the model. Compatible and incompatible virtual surgeries correspond to changes in the pulling directions of the muscles, defined according to whether the generation of force is compatible or not with the model's initial muscle synergies. The virtual surgeries were computed following the procedure described in [8], utilizing the environment matrix $H$, its null space (the subspace of the muscle activity space that does not affect the generated force) and the initial muscle synergies matrix $W$. From these, we derived the subspace bases necessary for the rotations in the muscle space: the basis $N_c$ of the subspace common to the synergies and the null space of $H$, the basis $W_{nc}$ of the synergy vectors not in the null space, and the basis $N_{nc}$ of the null space vectors not generated by synergy combinations. These subspaces were then used to calculate the rotation matrices for the surgeries: the compatible rotation matrix $T_c$ rotates a vector $w$ in the span of $W_{nc}$ onto a second vector $w'$ in the same subspace, while the incompatible rotation matrix $T_i$ rotates a vector

$w$ in the span of $\boldsymbol{W_{nc}}$ onto a vector $\boldsymbol{n}$ in the span of $\boldsymbol{N_{nc}}$. The angle of the compatible rotation was adjusted so that both compatible and incompatible virtual surgeries would have a similar "index of difficulty", defined as the average change in the muscle activity across muscles and force targets required to perform the task after the surgeries [8], calculated as:

$$I_{\text{diff}} = \sum_{i=1}^{M} \sum_{k=1}^{8} \left| m_{ik} - m'_{ik} \right|,$$

(9)

where $m_{ik}$ is the activity of the $i$th muscle for the $k$th target before the perturbation and $m'_{ik}$ is the same during the perturbation. The vectors $\boldsymbol{m_k}$ and $\boldsymbol{m'_k}$ are computed as the minimum-norm, nonnegative solution to the equations $\boldsymbol{f_k^*} = \boldsymbol{Hm_k}$ and $\boldsymbol{f_k^*} = \boldsymbol{H' m'_k}$, respectively (with $\boldsymbol{H'}$ being the perturbed environment matrix).

The effects of the virtual surgeries on baseline muscle synergies in representative simulations are illustrated in Fig 2A. It can be noted that compatible surgeries do not require changes to the muscle synergies in the model, but only adjustments in their relative recruitment across different force directions: the forces generated by the synergies (*black arrows*) are altered by the perturbation, yet they still span the entire force space. In contrast, incompatible surgeries require learning of new patterns of muscle activity, as the perturbed synergy forces are aligned in a single direction and do not span the force space.

Each simulated experiment started with the initialization of the model, followed by a baseline training without any perturbation for a total of 324 cycles, where each cycle consisted of eight trials—one trial for each of the eight defined targets presented in random order. During each trial, the learning rules described in the Results section were applied to update the matrices $\boldsymbol{Z}$, $\boldsymbol{W}$, and $\hat{\boldsymbol{H}}$ of the model. After the initialization, each simulated experiment was trained in the baseline condition for 36 cycles, followed by 36 cycles of perturbation (matching the number of trials used in the human experiments reported in [8]), and finally by 36 cycles of washout (again in the baseline condition). Within a single model initialization, the data for each trial were averaged across four repetitions of the simulation, to attenuate the effect of the motor noise added to the motor commands at every trial. All reported results (except individual reaching trials) were obtained by averaging, at every cycle, the data across 16 random initializations of the model. Each initialization used its own environment matrix $\boldsymbol{H}$ and muscle synergies $\boldsymbol{W}$. Because of this, each initialization also had different pairs of compatible and incompatible virtual surgeries. This approach was intended to mimic an experiment with multiple participants, and 16 initializations were deemed sufficient to illustrate the model's behavior (for comparison, the virtual surgeries results reported in [8] were obtained from eight participants).

We conducted three sets of simulations under different combinations of model parameters: the learning rates of the control policy $\eta_Z$, of the muscle synergies $\eta_W$, and of the forward model $\eta_{\hat{H}}$, as well as the regularization weights for the control policy $\lambda_Z$ and for the muscle synergies $\lambda_W$. The three simulations, along with the parameters investigated in each, are summarized in Table 1.

Simulation 1 tested the overall model and assessed the effects of updating the control policy and the muscle synergies. To this end, we set two possible values for the learning rate of the control policy $\eta_Z$ and of the muscle synergies $\eta_W$, resulting in three combinations of parameters: $\{(0.05, 0), (0.05, 0.05), (0, 0.05)\}$. The regularization weights of the control policy, $\lambda_Z$, and of the muscle synergies, $\lambda_W$, were set to 1% of the learning rate of each respective component (i.e., a 100:1 ratio of the weight for minimizing the task error versus regularizing the component in the cost function), which with a learning rate of 0.05 corresponds to a value of $5 \cdot 10^{-4}$, with the same order of magnitude as used in other simulation studies [22]. Regularization weights 4 times larger were also tested, resulting in motor commands with smaller norms and larger task errors, reflecting the trade-off between minimizing the error and minimizing the norm of the model components of the model. The learning rate of the forward model $\eta_{\hat{H}}$ was fixed at 0.25. This value, along with the combination $(0.05, 0.05)$ for the learning rates of the control policy and of the muscle synergies, respectively, was chosen based on an initial sensitivity analysis in which five different values for each parameter were tested across eight different model

initializations. This combination of parameters qualitatively reproduced the experimental results of [8] in terms of differences, between compatible and incompatible virtual surgeries, in force direction error and in reconstruction quality of the muscle activity using the original muscle synergies. Fig D in S4 Text illustrates the change in reconstruction quality ($R^2$) of the muscle activity using the original muscle synergies during this sensitivity analysis, only for the incompatible surgery perturbation, separately for each combination of learning rates, showing that the chosen parameters produced the observed decrease in $R^2$. Simulation 1 thus allowed us to examine the relative contribution of updating of each model components to overall learning. All parameter combinations were repeated across the 16 different initializations of the model.

In Simulation 2, we investigated the effect of the learning of the forward model $\hat{H}$ in the overall learning process. We tested two different learning rates $\eta_{\hat{H}}$, in $\{0, 0.25\}$, and a third condition in which the forward model was "ideal", i.e., it was equal to the perturbed environment matrix $H$, and to the unperturbed environment matrix during the baseline and washout. This third condition corresponds to instantaneous learning of the forward model, or *a priori* knowledge of the environment, as assumed in previous studies [22,23]. In this simulation, the learning rates of the control policy $\eta_Z$ and of the muscle synergies $\eta_W$ were both fixed at 0.05, and the regularization weights of each component were set to 1% of their respective learning rates, as in Simulation 1.

In Simulation 3, we investigated the effect of the regularization on the learning process. The learning rates of the control policy $\eta_Z$ and of the muscle synergies $\eta_W$ were both fixed at 0.05 and the learning rate of the forward model $\eta_{\hat{H}}$ was fixed at 0.25. Both regularization weights $\lambda_Z$ and $\lambda_W$ were set at either 1% of the respective learning rate or 0, allowing us to compare learning with and without regularization.

## 4.2 Performance metrics

Across all simulations, we quantified model performance using several metrics, calculated for each cycle of 8 trials. In each trial, the model generated force toward one of eight different force targets, with all eight targets presented once per cycle in random order. The first two metrics, *force direction error* and the *force magnitude error*, assessed the accuracy of the reaching movement. The *force direction error* was defined as the unsigned angle (in degrees) between two vectors: one from the center of the force space to the force target, and the other from the center of the force space to the force executed by the model. The *force magnitude error* was defined as the norm of the difference between the force target and the executed force (in arbitrary units). We note that, as our model generates feedforward muscle activity and forces, the *force direction error* corresponds to the angular error of the initial movement (i.e., the feedforward component of the movement) in human experimental data (referred to as *initial angle error* in Fig 2B). In contrast, the *force magnitude error* has no direct analogue in the human data, as a magnitude error cannot be clearly defined for the initial movement.

Two additional metrics were computed to characterize the muscle activity generated by the model. First, the *reconstruction quality of the muscle activity using the original muscle synergies* was evaluated. For each cycle, nonnegative linear least-squares optimization (MATLAB® function *lsqnonneg*) was used to find the nonnegative synergy recruitment coefficients that, when multiplied by the original muscle synergies of the simulation, minimized the squared norm of the difference with the muscle activity measured during that cycle. We then calculated the fraction of total variation of the reconstructed muscle activity explained by the synergies ($R^2$) across the trials of each cycle. The second metric was the *norm of the muscle activity*, computed as the average norm of the eight vectors of muscle activity generated in a cycle.

In Simulation 1, for the condition in which both the control policy and the synergies had non-zero learning rates, we also analyzed the norm of the muscle activity in different subspaces of the muscle activity space, to analyze how the usage of different dimensions changes throughout learning. Specifically, we computed the norm of the projection of the muscle activity in the *baseline task space* (the subspace of the muscle activity space that affects the force generated in the baseline [unperturbed] task, generated with the span of the transpose of the baseline environment matrix $H$), in *the null space* (the subspace which results in a zero change in the force executed in the environment $H$), in the $N_c$ *space* (the common

[compatible] subspace between the null space and the baseline muscle synergies), and in the $N_{nc}$ space (the subspace of the null space not intersecting [not compatible with] the baseline muscle synergies). For each subspace, the muscle activity vectors were projected, and their norms were calculated. This metric was also computed in Simulation 2, under the condition of an ideal forward model, with results presented in Fig C in S4 Text.

We computed two additional metrics in Simulation 1 to better assess changes in the structure of the muscle synergies during the perturbations. The first metric, the *area of the convex hull of the synergy forces*, represents the overall span of the forces associated with the muscle synergies within a given task environment. It was evaluated in both the baseline and the incompatible task spaces, including for simulations not subjected to the incompatible surgery. Following the procedure described in [36], the synergy vectors were first normalized to unit norm, then projected into the task environment. The area of the convex hull of the synergy forces was calculated and scaled by the area of the convex hull of the forces spanned by the individual muscles in that environment. The second metric, the *principal angle* between the synergies and the vectors $w$, $w'$, and $n$, quantifies the alignment of the subspace spanned by the synergies in the muscle activity space with the three vectors used to define the virtual surgeries. The *principal angles* were computed using the numerical procedure described in [71]. Both metrics were evaluated using the muscle synergy matrix saved at every three cycles, which corresponds to one block (both in our simulations and in experiments with human participants [8,10,36]).

In Simulation 1, under the parameter combination of a non-zero learning rate for both control policy and synergies, we also calculated the *force prediction error* of the forward model. This was defined as the norm of the difference between the force executed by the model and the force predicted by the forward model (in arbitrary units), evaluated for the motor command generated when reaching to each force target. The results of this analysis are presented in Fig A in S4 Text.

## 4.3 Statistical analysis

We analyzed the simulated data by fitting generalized linear mixed-effects models to the data at the last cycle (or block, for data saved at every three cycles) of the perturbation phase. The dependent variables were: force direction error, force magnitude error, fraction of total variation of the reconstructed muscle activity explained by the original synergies ($R^2$), norm of muscle activity (both in the entire muscle activity space and in each subspace). Random effects included the ID of each model initialization. In Simulations 1 and 2, independent variables were the perturbation type (rotation, compatible surgery, or incompatible surgery), the model parameter combination, and their interaction. For Simulation 1, additional models were fitted using as dependent variables: the norm of the muscle activity in different subspaces, the area of the convex hull of the synergy forces in the baseline and incompatible task spaces, and the principal angles between the synergies and three vectors in the muscle activity space. These were evaluated only for the model parameter combination where both the control policy and the muscle synergies were updated. In Simulation 3, we verified that only the model for the norm of muscle activity showed a significant effect of the interaction term; for all the other metrics, models were fitted without the interaction term. Post-hoc analyses were performed using F-tests on the fitted models using different contrasts, and Bonferroni-corrected *p*-values were computed. In Simulation 1, contrasts were defined across all perturbations and within each model parameter combination (for models of the norm of the muscle activity in different subspaces, contrasts were defined across all perturbations). In Simulation 2, and for the motor command norm in Simulation 3 (which included an interaction between perturbation type and model parameter combination), contrasts were defined across the different parameter combinations within each perturbation types.

## 4.4 Additional simulations

In the three simulations that we present in the manuscript, we demonstrate how the different components of our computational model contribute to its overall learning process. However, the model exhibits additional emergent properties that are not fully captured by these three simulations. Therefore, we conducted three additional simulations where we investigate features of the model not covered in the main text. A detailed description of these simulations and their results is provided

in S2 Text. In the simulation in Section A of S2 Text, we test the adaptation to incompatible surgeries when the model has a large number of fixed synergies, using either the "ground truth" synergies or an estimate of the model's synergies to calculate the surgery. The simulation in Section B of S2 Text examines training under incompatible surgeries followed by further training under compatible surgeries, and vice versa, to assess persistent changes in the model's behavior after adaptation. The simulation in Section C of S2 Text simulates the model without signal-dependent motor noise, to evaluate how motor noise affects the reorganization of the muscle activity patterns and the decrease of the force error.

## Supporting information

**S1 Text. Derivations of update equations of the computational model.**
(PDF)

**S2 Text. Additional simulations of the computational model.**
(PDF)

**S3 Text. Tables with results of statistical tests performed on data from simulations.**
(PDF)

**S4 Text. Supplementary figures from the simulations in the manuscript.**
(PDF)

**S1 Dataset. MATLAB scripts and processed data to reproduce figures and analyses from the simulations in the manuscript.**
(ZIP)

## Author contributions

**Conceptualization:** Lucas Rebelo Dal'Bello, Denise Jennifer Berger, Daniele Borzelli, Etienne Burdet, Andrea d'Avella.

**Data curation:** Lucas Rebelo Dal'Bello.

**Formal analysis:** Lucas Rebelo Dal'Bello, Etienne Burdet, Andrea d'Avella.

**Funding acquisition:** Denise Jennifer Berger, Andrea d'Avella.

**Investigation:** Lucas Rebelo Dal'Bello.

**Methodology:** Lucas Rebelo Dal'Bello.

**Project administration:** Andrea d'Avella.

**Software:** Lucas Rebelo Dal'Bello, Andrea d'Avella.

**Supervision:** Andrea d'Avella.

**Validation:** Lucas Rebelo Dal'Bello.

**Visualization:** Lucas Rebelo Dal'Bello.

**Writing – original draft:** Lucas Rebelo Dal'Bello.

**Writing – review & editing:** Denise Jennifer Berger, Daniele Borzelli, Etienne Burdet, Andrea d'Avella.

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
