## [Decision Letter · Decision Letter 0]

6 May 2025

PCOMPBIOL-D-25-00179

A modular architecture for trial-by-trial learning of redundant muscle activity patterns in novel sensorimotor tasks

PLOS Computational Biology

Dear Dr. Rebelo Dal'Bello,

Thank you for submitting your manuscript to PLOS Computational Biology. After careful consideration, we feel that it has merit but does not fully meet PLOS Computational Biology's publication criteria as it currently stands. Therefore, we invite you to submit a revised version of the manuscript that addresses the points raised during the review process.

Please submit your revised manuscript within 60 days Jul 06 2025 11:59PM. If you will need more time than this to complete your revisions, please reply to this message or contact the journal office at ploscompbiol@plos.org. Please include the following items when submitting your revised manuscript:

We look forward to receiving your revised manuscript.

Kind regards,

Tomohiko Takei

Guest Editor

PLOS Computational Biology

Andrea E. Martin

Section Editor

PLOS Computational Biology

**Additional Editor Comments :**

The three reviewers recognize the significance of this study, which aims to integrate three learning components—control policy, synergy structure, and internal forward model—within the virtual surgery paradigm. They also highlight the need for more detailed presentation of the results and a deeper discussion of their implications. A key recommendation is to include a discussion comparing the present findings with previous and alternative models. Please use this feedback to revise the manuscript.

**Journal Requirements:**

1) Please upload all main figures as separate Figure files in .tif or .eps format. For more information about how to convert and format your figure files please see our guidelines:

2) We have noticed that you have uploaded Supporting Information files, but you have not included a complete list of legends. Please add a full list of legends for all the Supporting Information files after the references list. We noted that S1 figure is labeled as S7 fig in the supporting information legend. Please correct the label of the figure.

**Reviewers' comments:**

Reviewer's Responses to Questions

Reviewer #1: The authors introduce a computational model of trial-by-trial generation and learning of muscle activity to test the modularity framework. They test the model on isometric force tasks with visuomotor rotations as well as compatible and incompatible virtual surgeries, consistent with prior experimental data from the group. They find that indeed their model that includes modularity explains the experimental virtual surgery data.

There are a few things that I think would improve clarity and readability of the manuscript

1. The introduction and discussion are on the longer side and could benefit from more succinct language. In the introduction, for example, there is good contextualizing of this work within the existing literature but does not state the findings from this paper. It could be clearer on what is new and novel about this model. A structure that describes the three simulations they present would be helpful (this information exists elsewhere in the paper but it’s buried in the methods). By doing so, it will be very clear what this work is adding to the existing literature and also easier to follow what the take-aways are.

2. There is a lack of statistics in the main body of the manuscript. The statistics are all in supplemental tables. It would be nice to include some indication of the statistically significant differences in the figures themselves. Especially because there are some comparisons that are statistically different but it is difficult to see differences in the figures themselves. Something like a bar along the top that shows the cycles for which there is a statistical difference would be helpful.

3. There are multiple statements that the models are consistent with the data of the compatible and incompatible surgeries. If possible, I think it could be very helpful to include the experimental data findings alongside the model outputs for direct comparison.

Minor concerns

1. I don’t understand the inclusion of the visuomotor rotation perturbation. If the model is meant to test de novo learning, visuomotor rotations are much more similar to adaptation than de novo learning. Also, there isn’t much difference between the modeling of visuomotor rotations and the compatible virtual surgeries. If it is included because visuomotor rotations are well-studied and thus is serves as a good benchmark for the model, that should be more explicitly stated.

2. In Figure 1, how is the back propagation of the force error through the forward model shown? There are many dashed lines but only one with an arrowhead on the dashed lines. Are those lines supposed to indicate the feedback?

3. I would benefit from more handholding in the descriptions of the Null space, Nc space and the Nnc space. I am not sure I understand what the “null space of the baseline task space” means. The description at Line 356-358 makes me think that the baseline task space and the null space are force spaces but they are described as “different subspaces in muscle activity space.”

4. The paragraph starting at Line 430 is difficult to follow. There are lots of pairwise comparisons that are summarized in a supplementary table. However, it is not easy to parse what the take-aways from all these comparisons are. Also, rather than just reporting that the reconstruction quality increases or decreases it would be helpful to describe what an increase/decrease means.

Specific comments

Line 367: should “force error” be “force magnitude error”?

Line 392: towards each of the 8 targets meaning one trial to each target? Or, eight trials to each target?

Figure 3: it would be helpful to label each panel with a number. This would simplify the callouts from the text.

Reviewer #2: 1. Summary of the manuscript

The authors presented the computational model of trial-by-trial learning of redundant muscle activity patterns during isometric force perturbation tasks under various perturbations such as visuomotor rotation and virtual surgery. A key feature of the proposed model is the assumption that the motor system operates through a modular control architecture. In this framework, muscle activity is generated by flexibly combining a small set of predefined spatial muscle synergies. These synergies are recruited through the control policy, which maps a desired force into specific combinations of synergies. The resulting muscle activation is then transformed into endpoint force via an internal model of the musculoskeletal system. Importantly, the model incorporates learning mechanisms in which the control policy, the muscle synergy structure, and the internal forward model are all updated based on sensory prediction errors. This allows the system to gradually refine the structure of its motor output and its internal understanding of how muscle activity generates force.

Through simulations, the authors show that successful adaptation, especially under incompatible virtual surgeries where the baseline synergies can no longer span the force space, requires both structural updates to W and Z. The model also highlights the role of regularization in effort minimization and reproduces several experimental phenomena, such as aftereffects and changes in muscle activity structure. Overall, the model offers a mechanistic explanation for how the motor system may learn and adapt complex motor tasks through a redundant body by updating all levels of modular architecture.

2. Comments

The authors begin by systematically examining how the behavior of the model changes depending on different parameter settings, including the learning rates and regularization strengths applied to various components. Following this, they demonstrate that to successfully reproduce key findings from previous virtual surgery experiments, it is necessary to allow learning to occur across all levels of modular architecture, including the control policy, the synergy structure, and the internal forward model.

One major comment is that the manuscript would benefit from more intuitive explanations of the model’s behavior under certain conditions.

* I would appreciate a clearer discussion of why, during the washout phase, the force magnitude error remains large when both the control policy and the synergy structure are updated and why this phenomenon does not occur when only the synergy structure is updated. A mechanistic explanation would help the reader better understand the implications of updating different layers of the modular architecture (line 449).

* The simulation results of this model are quite intuitive: when force cannot be generated in the task space, the system appears to redirect activity into the task null space. However, I am struggling to fully understand why such a simple gradient-based model, updating the synergy matrix W using a force-to-muscle mapping derived from the internal forward model ^ ⊤, can produce such a task-space to null-space reallocation of activity. Given that the control policy Z is also updated based on a similar gradient derived from the same error signal, I would like to better understand how this redistribution emerges. Does this transition reflect a division of roles between the two parallel learning pathways (i.e., synergy structure vs control policy)? More specifically, how does the model determine that the task space is no longer sufficient and that expanding into the null space is necessary? An explicit discussion of the complementary or competing contributions of W and Z during this process would help clarify the source of the observed structural reorganization in the muscle activity space. (Line 530)

Another concern is that the model appears to be configured in a way that closely reproduces known experimental outcomes, but it is less clear whether it provides new insights or explanatory mechanisms beyond what is already known. In other words, rather than clarifying the underlying principles of motor adaptation, the model may simply be encoding prior empirical findings into its structure.

* One promising direction that could be further explored or discussed is how the model might help explain individual differences in motor adaptation. The authors demonstrate that the model's behavior changes substantially depending on the parameter settings—such as the relative learning rates of the control policy and synergy structure or the accuracy of the internal forward model (Fig. 3-5). These factors may map meaningfully onto inter-individual variability observed in experimental data. For example, participants with greater baseline variability might correspond to a less accurate internal model, while others with more consistent baseline performance may reflect more refined internal representations. If the authors have access to such participant-level data showing that (e.g.,) the variability in the baseline trial is correlated with the learning speed in the incompatible perturbation, it would be highly valuable to explore whether individual differences in adaptation can be predicted by the model under different parameter combinations. Even if such data are not currently available, we believe that acknowledging this possibility in the discussion would highlight the model’s potential for generalization and translational relevance.

* Another point worth expanding is the distinction between modifying existing muscle synergies versus increasing the dimensionality of control—a key theoretical question in motor control that is briefly mentioned as a limitation (Line 749). Their proposed model allows for structural updates to the synergy matrix W, which could be interpreted as reorganizing existing synergies. I wonder whether it is good to compare their proposed model with alternative models, particularly ones that assume fixed muscle synergies but increased control dimensionality as a means of adaptation. For example, I would argue that the large decrease in R² through the incompatible perturbation observed in the simulation of the proposed model supports the necessity of structural change in synergy composition rather than merely increasing control dimensionality. If the adaptation could be accomplished simply by recruiting additional dimensions, one would expect R² to remain relatively stable (Line 446). In contrast, while the authors interpret the larger decrease in R² observed experimentally during incompatible surgeries as a potential sign of higher-dimensional control strategies, another possibility is that both structural reorganization and increased control dimensionality are engaged simultaneously in adaptation. It might be worth clarifying this point and reconsidering the interpretation considering this alternative perspective (Line 449).

* Another interesting implication of the current model is that residual changes in the structure of muscle activity after adaptation to an incompatible perturbation—such as the incomplete recovery of R² or the persistently elevated muscle activity norm during washout—may influence subsequent motor learning.

For instance, Berger et al. (2023) showed that prior exposure to an incompatible virtual surgery affected the adaptation to a subsequent compatible surgery, resulting in reduced reconstruction quality and increased reaction times—whereas the reverse order had no such effect. This raises the question: Can the current model reproduce such history-dependent learning asymmetries? If so, demonstrating this would greatly enhance the explanatory power of the model (Line 463).

Lastly, as the model components (e.g., synergy structure, control policy, internal forward model) are meant to reflect functional modules in motor control, it would be valuable to include at least a brief discussion of how these elements might relate to neuroanatomical substrates. While the simulations reproduce a range of experimental findings, it remains somewhat unclear why this architectural design is necessary or advantageous beyond its ability to match known data. I wonder whether this architecture is robust to environmental changes or motor noise.

3. Conclusion

Overall, I find this work a valuable contribution to the field of computational motor control. That said, I believe that the manuscript would benefit from further clarification of the model’s mechanisms and assumptions, as well as a deeper discussion of its limitations and comparison with other hypotheses, as these revisions are likely to substantially improve the clarity and significance of the work.

Reviewer #3: This study proposes a computational model of motor learning in tasks involving isometric force generation with the arm. In particular, the study models learning in 3 different tasks: a visuomotor rotation (a transformation in force space), a compatible virtual surgery, and an incompatible virtual surgery (transformations in muscle space). Here, the brain-body system is modeled as a system with 4 main components: 1. A feedforward controller that transforms a desired arm end-point force into a low-dimensional motor command, 2. Muscle synergy modules which transform the low-dimensional motor command into the muscle activations of a set of arm muscles, 3. An arm, represented as a linear mapping between muscle activations and end-point forces, and 4. A forward model of the arm that estimates produced forces given an efferent copy of the muscle activations. The forward model of the arm is used to transform errors in the force space into errors in the motor command space so that the feedforward controller and the muscle synergies can be adjusted to minimize errors in the 3 isometric tasks. The novelty of the study corresponds to the inclusion of learnable muscle synergies and forward model of the arm, which had not been included simultaneously in previous modeling studies. By assuming that learning in the feedforward controller is faster than learning in the muscle synergy structure, and that learning of the forward model of the arm occurs slowly, the study qualitatively replicates task behavior and muscle activity observed in previous experiments.

The study is interesting and gives a complementary view of learning in tasks that require the acquisition of new spatial muscle activation patterns, but there are some major issues that need to be addressed:

1. Based on the equations that define the model (plant and gradients), muscle activations do not seem to be restricted to be non-negative. This is very important, as negative activation of muscles would result in muscles that produce pushing forces, which is not biologically feasible. As a workaround, it seems like the feedforward controller was initialized in a region of parameter space that tends to produce non-negative muscle activations (Line 240). However, there is no guarantee that this will remain to be the case during learning, especially for the incompatible virtual surgery, where the synergy weights don’t seem to be restricted either. It is difficult to be confident in the interpretability of the results without guaranteeing non-negative muscle activations.

2. The addition of motor noise in muscle activity has no bearing in the learning process. That is, the noise does not inform any aspect of the learning algorithm as in previous studies of this task (DalBello and Izawa, 2022). Performance in Fig 2A suggests that noise is quite large. At best, the inclusion of noise adds some biological plausibility to the model, but its relevance in the current study is unclear. At worst, a) It might produce biologically implausible results, and b) It obscures the results of the simulations, increasing the difficulty of result interpretation.

a. Because the model does not guarantee non-negative muscle activations, in some cases, noise may pull down the motor command to be negative.

b. The R^2 reconstructions shown in Fig. 3 suggest that noise may affect this metric in an important way. In the case where only the control policy is updated (synergies are fixed), there is a drop in the R^2 metric for the rotation and compatible tasks. All motor commands are produced by the fixed synergies, so the only reason that I can think for this drop is motor noise. At the same time of this drop in R^2, there is an increase in the size of the motor command. Because the noise is signal dependent, the larger the motor command, the larger the magnitude of the noise. Therefore, there is a strong possibility that noise might be a confounding factor in the dips seen in R^2. This is especially worrisome for the case of the incompatible surgery when both control policy and synergies are updated. The motor command in this case becomes very large. Therefore, it is difficult to attribute this drop in R^2 exclusively to the model configuration (update of control policy and synergies). Given that this drop in R^2 is the main result of the study, this issue needs clarification. Therefore, I recommend running a round of simulations without any motor noise.

3. Even though the learning of the forward model is a central component in the current model, no learning curves of how the forward model is learned are shown. That is, the time course of the prediction error throughout the task is not available. It would be insightful to see the time scale of learning in this model compared to the time scale of learning in the overall task. For example, the results of Pierella et al. 2019 suggest that the forward model is learned significantly faster than the control policy. This would shed light on how the selection of the learning rate parameters in the current model affect the results of the simulation. It is mentioned that the specific values of the parameters were selected through an analysis to qualitatively reproduce experimental results (Line 310-316). It would be important to discuss more granularly under what conditions these results hold.

4. The presentation of the study emphasizes the contrast between the merits of previous models and the current model, and suggests the current model is superior. Here, the cause of the difference between learning speed in compatible and incompatible synergies is attributed to the difference in learning rates between the control policy and the muscle synergies in conjunction with slow learning of an inverse model. Other models have attributed this difference to:

a. The structure of exploration noise and its interaction with both types of surgeries (task-relevant and non-task relevant noise) (DalBello and Izawa, 2022). In this case, the strength of the current model is argued to be the inclusion of modularity in the model, but it seems trivial to include modularity in the DalBello and Izawa study to get similar results, so the comparison seems quite superficial.

b. The structure of the error surface in the learning parameter space (Barradas et al, 2024). In this case, the strength of the current model is argued to be the inclusion of learnable synergies and forward model. However, an analysis of the Hessian of the error surface in the current model is not provided. Therefore, it is not clear what the predictions of the Barradas et al study are in the current architecture. The goal of the Barradas et al paper is to show the influence of the structure of the parameter space on motor learning, and not necessarily to define an architecture for learning the virtual surgery task.

The bottom line here is that in the real world there are likely many parallel factors that bring about the difference in learning rates between the compatible and incompatible virtual surgeries and other tasks. Therefore, rather than presenting a false dilemma between “competing” explanations, it would be more insightful to compare how these explanations interact with each other.

5. The extent of the predictions of the model are limited to previously observed data. The current model could be more impactful if it could be used to make predictions for future experiments in the virtual surgery and other tasks. Perhaps the results of the projections of the motor commands onto the different task and synergy spaces could be a way to do this.

Minor comments:

1. The paper does not follow the typical format of the PLOS Computational Biology journal (Order of sections).

2. The structure of the forward model is assumed to be known a priori. What would be the influence of using a more realistic structure? How do the results depend on the choice of forward model mapping? For example: What if the forward model is a general function approximator like the control policy network? What if the forward model maps synergy activation coefficients to predicted forces instead of muscle activations to predicted forces?

3. Awkward phrasing in some parts. For example:

a. Line 173-176: which assumed that the “knowledge” of the environment is “known” a priori

b. Line 175: visualized or realized?

c. Line 235: circle distant 0.5 from the

d. Line 251: surgeries were initialized or defined?

e. Line 303: “of and with synergies”

f. Line 342: Awkward definition of force direction error

4. In Fig. 1 it is not immediately clear what the figures above the diagram of the model are. They are the H mapping, the W matrix and…

5. Given that the H and W matrices were generated artificially across 16 different initializations, why not make many more initializations?

6. Line 263: Maybe include Index of difficult equation in supplementary material?

7. Fig. 2 contains results but it is in the Methods section.

8. It sounds artificial to introduce the regularization term as such, when it can be completely derived from the gradient of the effort defined as the sum of squared muscle activations.

9. It is more natural to justify the value of the regularization as a trade-off between error and effort units rather than as a percentage of the learning rate.

10. Line 354 – 361: The purpose of analyzing the motor command projection in different spaces should be explained here.

11. It would be nice to see a comparison with the previously published experimental results to confirm that the difference in learning speed of compatible and incompatible surgeries is relatively similar to the simulations.

12. It is unclear why statistical comparisons are performed between the visuomotor rotation and the virtual surgeries. What knowledge is gained from this?

13. Fig 3: The aftereffect for the error in the simulations appear much larger than what has been found experimentally. I could not find the actual value in the Berger et al 2013 paper, as the first washout block is averaged across many trials.

14. Line 476 – 484: Discussion material, not results. This is true in other paragraphs too.

15. Fig. 4: It would be useful to clarify what the projection onto the baseline task is.

16. Fig. 5: Why doesn’t the error increase at washout in the ideal forward model case for the incompatible surgery?

17. Please include a README file with instructions to run the Matlab scripts.

**Figure resubmission:**
---

## [Decision Letter · Decision Letter 1]

7 Sep 2025

PCOMPBIOL-D-25-00179R1

A modular architecture for trial-by-trial learning of redundant muscle activity patterns in novel sensorimotor tasks

PLOS Computational Biology

Dear Dr. Rebelo Dal'Bello,

Thank you for submitting your manuscript to PLOS Computational Biology. After careful consideration, we feel that it has merit but does not fully meet PLOS Computational Biology's publication criteria as it currently stands. Therefore, we invite you to submit a revised version of the manuscript that addresses the points raised during the review process.

Please submit your revised manuscript within 60 days Nov 07 2025 11:59PM. If you will need more time than this to complete your revisions, please reply to this message or contact the journal office at ploscompbiol@plos.org. Please include the following items when submitting your revised manuscript:

We look forward to receiving your revised manuscript.

Kind regards,

Tomohiko Takei

Guest Editor

PLOS Computational Biology

Andrea E. Martin

Section Editor

PLOS Computational Biology

**Additional Editor Comments:**

The reviewers acknowledge that your revision has substantially improved the clarity and strength of the manuscript. In particular, the additional simulations have helped substantiate several of the main claims. However, the reviewers have also emphasized the necessity to more explicitly elaborate on the differences between the model predictions and the experimental observations. Accordingly, I would like to ask you to revise the manuscript to provide a clearer discussion of these discrepancies and, where possible, to clarify potential mechanisms that may account for them.

**Journal Requirements:**

1) We have noticed that you have uploaded Supporting Information files, but you have not included a complete list of legends. Please add a full list of legends for your Supporting Information files (processed data.zip) after the references list.

**Reviewers' comments:**

Reviewer's Responses to Questions

Reviewer #1: I thank the authors for their careful responses to my feedback and their revisions to the manuscript. They have largely addressed my concerns. I have only a few remaining comments:

1. The inclusion of the experimental data figure panels are helpful. It would be even more helpful if the y-axes were the same scale. In B and C, is ‘angular error’ the same as ‘force direction error’? If so, consistent labels would be clearer.

2. Line 325-327: “…under all 3 combinations of learning rates.” Is the comparison really between different learning rates or is the comparison between different learning ‘modes’, i.e., updating synergies vs the forward model?

3. Line 329-332: “While a decrease in the reconstruction quality suggests that the structure of the underlying muscle synergies have changed compared to the baseline, this is not possible in the simulations without the update of the muscle synergies.” Is this by construction? Also, it is unclear if this statement applies to the rotations and compatible surgeries or just the incompatible surgeries.

4. I found the statistics indicated in Figure 5 to be a little confusing. I think it is because it looks like the asterisk is being labeled as incompatible rather than clarifying which conditions are being compared.

5. Some of the sentences, especially in the discussion, are very long. Because of this, I found parts difficult to follow

Typos or text comments:

1. I think there is a missing word in the sentence line 295-296.

2. Line 300: missing an ‘and’ between Z and W

3. Figure 3 caption, line 314: ‘Each panel…’ should be ‘Each row…’

4. Line 361: “We also observed that both force errors did not return…” -> “We also observed that neither force error returned…”

5. Line 373: “This behavior is replicated in an additional…”

6. Line 706: What does “larger” forward model mean? Closer to ideal? Higher dimensional?

Reviewer #2: I appreciate the extensive revisions the authors have made. The revised manuscript provides a much clearer explanation of the hierarchical learning framework and its implications for motor adaptation. The authors have successfully addressed most of my previous comments, and the inclusion of additional simulations has helped substantiate several key claims. Overall, I find the manuscript substantially improved and believe it now presents a compelling mechanistic account of trial-by-trial motor learning in a modular system.

That said, I have three remaining comments that I encourage the authors to consider to further clarify and strengthen their contribution:

Comment

Clarifying the Temporal Profile of Muscle Activity Redistribution (Figure 4):

In the discussion surrounding Figure 4 (lines 434–472), the authors describe a progression in which muscle activity, particularly under the incompatible surgery condition, initially increases in the baseline task space and is later redistributed into the N_nc space. While this explanation is informative and plausible, it gives the impression of a sequential transition: that is, the baseline component increases first, followed by a delayed rise in N_nc activity.

However, upon close inspection of Figure 4, the time courses of the baseline and N_nc projections appear to rise concurrently during the initial learning phase. The more striking difference lies in the sustained elevation of the N_nc norm, whereas the baseline component tends to plateau. This suggests that both subspaces may be recruited in parallel, and that N_nc reflects more lasting structural adaptation.

If the authors intend to argue for a sequential transition from baseline to N_nc, we recommend including a quantitative comparison of onset latencies or rise times between these components to support this interpretation. Alternatively, we suggest revising the text to clarify that both subspaces are likely co-recruited early in learning, with N_nc becoming more prominent over time. Additionally, since the incompatible condition demonstrates the clearest structural reorganization, the authors might consider including a consolidated figure overlaying the time courses of the baseline, N_c, and N_nc norms for this condition. This would facilitate a more direct comparison and enhance the clarity of the redistribution dynamics.

Linking Findings to Broader Principles in Motor Learning:

The proposed model provides a sophisticated and compelling explanation of motor adaptation under virtual surgery conditions. However, its significance could be further enhanced by more clearly linking it to general principles and findings in the motor learning literature.

For example, the observed reduction in the norm of muscle activity through regularization is consistent with well-established findings that prolonged practice leads to a reduction in effort or control cost (Huang et al., J Neurosci, 2012). Additionally, it is known that learning is often impaired in inconsistent environments (Castro et al., Curr Biol 2014). Discussing how the current model accounts for such findings—or how it might behave under such conditions—would help position it more broadly within motor learning theory.

By explicitly highlighting these connections in the Discussion, the authors could more clearly situate the model within the broader context of sensorimotor learning research.

Probing the Role and Implementation of Noise in the Model

Furthermore, the model’s finding that forward model learning is facilitated by motor noise is consistent with the results of Wu et al. (2014), which showed that higher baseline variability predicts faster learning.

However, in the current implementation, noise is added only at the motor output level, and not to internal representations or prediction errors. Providing a rationale for this modeling choice would help deepen readers’ understanding. Additionally, discussing how the model’s behavior or predictions might change if noise were introduced at other levels—such as the forward model or internal state estimation—would strengthen the theoretical generality and applicability of the model.

Conclusion

Overall, this revised manuscript represents a significant step forward. The authors have responded thoughtfully to prior comments and clarified many key mechanisms in the model. The proposed modular learning framework is compelling and well-grounded in experimental observations. That said, addressing the remaining points—particularly the temporal interpretation of redistribution dynamics and the model’s relation to broader principles in motor learning—will help enhance both clarity and theoretical reach. I believe these refinements will make the paper more informative and impactful for the motor control and computational neuroscience communities.

Reviewer #3: Thank you for your thorough response to my comments. However, I think that the claim the study makes of providing a good fit to the experimental data is not fully supported. First, no formal fits to the data are provided or quantified, so it’s hard to claim a good fit to the data. Second, even though the model can produce dips in the R^2 metric during exposure to the incompatible surgery, the magnitudes of these dips remain considerably smaller than those observed experimentally, and only a fraction of the total dip in the model seems to be attributable to the proposed computational framework, with the other component coming from motor noise. See my response to comment 2 for a more in-depth discussion of the issue. I would recommend softening the claims of the paper, or performing additional supporting analyses. Additionally, I am including below other observations to some of the authors’ responses. I use the same enumeration as in my original review.

Main comments

1. This explanation helps to clarify the handling of the non-negativity constraint on muscle activations. Setting any resulting negative weights in the Z and W matrices to zero during learning would indeed ensure that muscle activations remain non-negative, provided that the network’s RBF input is also non-negative, which is guaranteed by their definition as Gaussian. However, a remaining small issue with this method is that the gradient component from these zero-weights could vanish, which could have some implications for learning. Could the authors report if this happened in the simulations, and how often if so?

2. After removing the motor noise, the R^2 value indeed drops noticeably during the incompatible surgery in comparison with the other conditions. I agree that this does show that the structure of the synergies W changes more during this model condition than in the other conditions. However, as the authors acknowledge (Line 350), the value of the dip in R^2 observed in the simulations is much smaller than that observed experimentally. Seeing as motor noise has a considerable contribution to the dip in R^2 (S2 Text), the size of the effect of control policy and synergy updates on R^2 predicted by the simulation appears now even smaller. Could the authors elaborate on why this difference with the experimental results is so large? The authors mention in the discussion section that cognitive processes might account for a portion of this additional drop in R^2 through the regulation of motor exploration. However, this would imply that the cognitive system can voluntarily produce explorative actions in muscle activation spaces outside of the spaces spanned by both the baseline and the newly learned synergies (if these actions are in the space of the newly learned synergies, the values of R^2 should not change with respect to the condition without this kind of explorative actions). What voluntary motor control processes would allow for this kind of motor exploration? To my knowledge, only a few plausible mechanisms come to mind: 1. Individual muscle control. 2. Production of high forces to engage signal-dependent motor noise that the system can somehow cognitively identify. 3. Engagement of additional muscle synergies that were not expressed in the baseline force task (i.e. synergies that produce forces out of plane). I think that more concrete hypotheses are needed, seeing as the update in muscle synergies by itself cannot account for the whole decrease in R^2.

Additionally, I now realize that these simulations offer a perfect opportunity to assess what the concrete changes in muscle synergy structure are after exposure to the incompatible surgery, and to relate them to the observed decreases in the R^2 metric. This is hard to perform experimentally as a large amount of stable data is needed to extract reliable muscle synergies. Therefore, extracting muscle synergies at the end of an experiment to directly compare them to the synergies before the experiment is hard to consider, and the R^2 metric is a clever way to assess the divergence of the muscle activation spaces. However, this model allows to perform this comparison directly. Informative outcomes like the similarity of the pre- and post- learning synergy sets, or the synergy forces spanning the horizontal plane can be studied, and correlated to the decreases in the R^2 metric. I apologize to the authors for not proposing this previously, but I think such an analysis would strengthen the study’s results.

3. I think it would be beneficial to include in the main text the explanation about the role of the exploration noise contributing to the acquisition of a richer forward model, leading to faster learning overall, as detailed in the last paragraph of the supplementary material. Otherwise, without any motivating context, the lack of explicit interaction of motor noise in the system’s learning equations may seem puzzling at first sight.

4. Thank you for including the figures showing the time course of force prediction error for the three model conditions. I think it makes sense that learning the forward model for the incompatible surgery has a faster component than the other two tasks. This is because the dimensionality of the incompatible task is reduced, so it is qualitatively “simpler” to learn (the force goes either in one direction or the opposite direction, at least at the beginning of learning). It then seems to reach a bottleneck. It would be interesting to see what happens if the model is allowed to continue learning the incompatible surgery. Can it totally eliminate the task error, and will the learning of the forward model be complete by then?

Regarding the sensitivity analysis for the learning rate of the forward model in Figure S4.C, from these figures it seems like the effect of the forward model learning rate on the size of the change of R^2 only has a marginal effect. That is, it seems to make the left tail of the shown distributions heavier, but the bulk of the simulations still show very little change (the means are quite close to zero). This may be a little misleading because these simulations presumably include many instances where learning of the control policy and synergies is very slow. This is evidenced from the fact that the mean change in the simulations (red line) is quite different from the mean of these histograms. Is there a way to visualize more directly the effect of the forward model learning rate? For example, by showing additional histograms when the other two learning rates (control policies and synergies) are fixed? Or subdividing each histogram bar into different color blocks indicating the combination of parameters used (there are only 8 combinations, right)?

5. Thank you for your perspective on my comment. I consider that both supplementary materials S2 and S4 contribute to a stronger analysis of the presented model and its relation with previous studies.

6. Thank you for providing the additional simulations supporting the Berger and d’Avella 2023 study. However, I consider that it is unclear what aspects of the model introduced here are essential to obtain this result. I suppose that this is because of the updateable weights of the muscle synergies. Because there is no memory component that would allow the system to return to the same baseline synergy structure after exposure to an incompatible surgery, it would be rather surprising if this experimental result could not be accounted for by the current model.

Minor comments:

1. No comment

2. It is a good compromise to include these points in the Discussion. However, I think it is important to emphasize there that the current model assumes that the mathematical structure of the forward model is the same as the structure of the muscle-force mapping (both are Mx2 matrices).

3. No comment

4. It’s a good idea to relate each of the insets to a block in the model. However, I think it would be illustrative to label the axes of the color maps (phi(f_x) and phi(f_y), correct?)

5. No comment.

6. No comment

7. No comment

8. I see. This is literally a regularization term as defined in machine learning. It may have a similar overall effect to minimization of effort, so no further comment.

9. No comment

10. No comment

11. It would be very helpful if the results in Figure 2 could have the same units as the experimental data (blocks vs cycles), or at least an axis that shows the correspondence between blocks and cycles.

12. The similarity between visuomotor rotations and compatible surgeries becomes evident from plots showing the forces in the plane associated to each synergy like those in Fig. 2A. The compatible surgery as defined in Berger et al 2013 will never mix up the distribution of synergy forces around the plane (the forces always stay in the same relative arrangement), so it effectively has a very similar effect to a VMR on these forces. It would be possible to create something like a mirror transformation to the baseline mapping, and this would still be compatible, but the procedure defined in Berger et al 2013 cannot produce such transformations (although it would be simple to do it manually). Therefore, the only differences between the compatible surgery and the VMR are small in the directions and magnitudes of the forces, so they should have very similar effects.

13. No comment

14. No comment

15. No comment

16. No comment

17. Thank you for including the README file. Unfortunately, I do not have access to the parallel computing toolbox, so I was not able to run the code. The README file states that the parallel computing toolbox is optional, but I don’t see a way to run the code without it. Is there an option in the script that initializes the simulations to activate/deactivate it?

Additional new comments:

The re-written sections of the paper need some proof-reading as there are plenty of typos.

Example: Line 405: “Fig. 2 illustrates the three types perturbations”

Line 417: “a modular architecture whose is slower…”

Line 469: “under different combinations of learning rates the model’s adaptive…”

Line 477: “when both Z W are updated”

Line 561: “This behavior is replicated an additional simulation…”

Line 484: Seems like a non sequitur? Because the incompatible virtual surgeries…

In summary, while the manuscript has improved substantially, I believe the central claims still need either stronger quantitative support or more cautious framing.

**Figure resubmission:**
---

## [Decision Letter · Decision Letter 2]

24 Dec 2025

PCOMPBIOL-D-25-00179R2

A modular architecture for trial-by-trial learning of redundant muscle activity patterns in novel sensorimotor tasks

PLOS Computational Biology

Dear Dr. Rebelo Dal'Bello,

Thank you for submitting your manuscript to PLOS Computational Biology. After careful consideration, we feel that it has merit but does not fully meet PLOS Computational Biology's publication criteria as it currently stands. Therefore, we invite you to submit a revised version of the manuscript that addresses the points raised during the review process.

We look forward to receiving your revised manuscript.

Kind regards,

Shlomi Haar, PhD

Academic Editor

PLOS Computational Biology

Andrea E. Martin

Section Editor

PLOS Computational Biology

**Reviewers' comments:**

Reviewer's Responses to Questions

**Comments to the Authors:**

Reviewer #1: I thank the authors for their thoughtful responses to reviewer feedback. My only remaining suggestion is that the manuscript would benefit from additional editing for clarity and concision. Several sentences are quite long and difficult to parse, and the discussion section has become rather lengthy in the process of addressing all reviewer comments. Streamlining these areas would improve readability.

Specific comments:

Line 345: “…this is not possible in the simulations without the update of the muscle synergies.” It may be more appropriate to say, “…this does not occur in the simulations without an update of the muscle synergies.” Please confirm that this wording accurately reflects the intended meaning.

Line 346-350: This sentence is quite long and difficult to parse. It would help to clarify what “observed here” (Line 347) refers to. Does “here” mean all panels in Fig. 3C or a specific panel? Also, there appears to be a typo in Line 350.

Fig. 6B: All y-axis labels should be consistent

Reviewer #2: Thank you for the thorough revisions. Although my own comments did not require substantial changes, I greatly appreciate the care with which you addressed the other reviewers’ points. These additions and clarifications have, in my view, significantly strengthened the validity and depth of the proposed model.

One of the most important improvements is the inclusion of new analyses that clarify how changes in muscle synergy structure unfold during learning. The convex-hull analysis of synergy forces and the principal-angle analysis between the synergies and the vectors defining the virtual surgeries together provide a more mechanistic explanation for the reduction in R² observed during incompatible surgeries. These analyses convincingly connect the model dynamics more directly to the experimental literature and enhance the credibility and interpretability of the simulation results. Importantly, they also reinforce the idea that this model is not merely a descriptive or reproducing framework, but one that offers insight into potential underlying mechanisms.

I also believe that the model has strong potential for further development. By linking the computational components to neural activity, individual differences, or cortical/subcortical learning mechanisms, future work could explore where and how such adaptive processes might be implemented in the brain. In this sense, the present study provides a valuable foundation for more integrative neuro-computational research.

My remaining comments are very minor and intended purely as optional suggestions to further clarify certain aspects of the manuscript. The paper is already strong as it stands, and these points are not essential for acceptance:

Recovery of the convex hull area (Fig. 5A):

The fact that the convex hull area does not fully recover after the incompatible surgery also made me think about whether this might contribute to the more pronounced decrease in R². If the synergy structure remains partially altered, the baseline muscle patterns may be less well reconstructed. A short comment on this possible relationship could further strengthen the interpretation.

Explanation of the vectors w, w′, and n (Fig. 5B):

I found the principal-angle analysis very informative, but I think the manuscript would benefit from a clearer conceptual explanation of what the three reference vectors (w, w′, n) represent. Providing a brief intuitive rationale for why these vectors were chosen, and how to interpret the angle changes with respect to each of them, would greatly help readers understand the significance of the observed synergy reorganization.

Number of model initializations (16 runs):

I noticed that the results are averaged across 16 model initializations. While this seems like a reasonable and standard choice, it may be helpful to briefly mention why this number was selected—for example, whether it reflects computational considerations or whether preliminary tests indicated that 16 runs were sufficient to stabilize the variance. This is only a minor point, and if there was no particular justification beyond practicality, a short statement to that effect would also be perfectly fine.

Regularization strength in Simulation 3:

I appreciate the comparison between the model with and without regularization. I was wondering whether the authors explored different strengths of the regularization term as well. Even a brief comment on whether varying the regularization weight changes the model’s behavior (or whether the effects remain qualitatively similar) would help clarify how sensitive the results are to this choice. Of course, if preliminary tests indicated that the strength has minimal influence, a short statement to that effect would already be helpful.

Reviewer #3: I would like to thank the authors for their efforts in addressing my concerns. I consider that the manuscript has become stronger after the revisions. I only have a couple of minor comments (the numbering follows my earlier comments), which require little or no action:

1. Thank you for the thorough response. This additional analysis suggests that the model is able to recover from gradient clipping, so the gradient does not end up vanishing. It’s interesting that the clipping is more pervasive in the incompatible surgery, suggesting that the computed gradients for the compatible surgery match the gradient of the linear network more closely than for the incompatible surgery. It would be interesting to explore the relationship between gradient clipping and the speed of learning, although this would be outside of the scope of the current study.

2. Thank you for elaborating on the possible mechanisms by which exploration with additional non-identified synergies may contribute to an increased drop in the R^2 metric, and for including the new set of simulations examining changes in the structure of muscle synergies. I would only suggest to more explicitly define the vectors w, w’ and n, as they are introduced somewhat abruptly, and might be difficult for readers to interpret at first.

3. I was able to run the provided code (do_sim_1.m and do_sim_2.m), but after the initialization stage indicated by the console output:

initialization 16/16

init 16/16, rot

init 16/16, comp

init 16/16, incomp

I encountered repeated instances of the following message:

Matrix size doesn't match, attempting to transpose matrix

The execution continued for some time, but given the repeated warnings and the long runtime, I stopped the process. It is not clear whether this behavior is expected. For reference, I ran the code on Matlab R2023a.

**Have the authors made all data and (if applicable) computational code underlying the findings in their manuscript fully available?**

The PLOS Data policy requires authors to make all data and code underlying the findings described in their manuscript fully available without restriction, with rare exception (please refer to the Data Availability Statement in the manuscript PDF file). The data and code should be provided as part of the manuscript or its supporting information, or deposited to a public repository. For example, in addition to summary statistics, the data points behind means, medians and variance measures should be available. If there are restrictions on publicly sharing data or code —e.g. participant privacy or use of data from a third party—those must be specified.requires authors to make all data and code underlying the findings described in their manuscript fully available without restriction, with rare exception (please refer to the Data Availability Statement in the manuscript PDF file). The data and code should be provided as part of the manuscript or its supporting information, or deposited to a public repository. For example, in addition to summary statistics, the data points behind means, medians and variance measures should be available. If there are restrictions on publicly sharing data or code —e.g. participant privacy or use of data from a third party—those must be specified.requires authors to make all data and code underlying the findings described in their manuscript fully available without restriction, with rare exception (please refer to the Data Availability Statement in the manuscript PDF file). The data and code should be provided as part of the manuscript or its supporting information, or deposited to a public repository. For example, in addition to summary statistics, the data points behind means, medians and variance measures should be available. If there are restrictions on publicly sharing data or code —e.g. participant privacy or use of data from a third party—those must be specified.requires authors to make all data and code underlying the findings described in their manuscript fully available without restriction, with rare exception (please refer to the Data Availability Statement in the manuscript PDF file). The data and code should be provided as part of the manuscript or its supporting information, or deposited to a public repository. For example, in addition to summary statistics, the data points behind means, medians and variance measures should be available. If there are restrictions on publicly sharing data or code —e.g. participant privacy or use of data from a third party—those must be specified.

Reviewer #1: Yes

Reviewer #2: Yes

Reviewer #3: Yes

PLOS authors have the option to publish the peer review history of their article (what does this mean?). If published, this will include your full peer review and any attached files.). If published, this will include your full peer review and any attached files.). If published, this will include your full peer review and any attached files.). If published, this will include your full peer review and any attached files.

...

Reviewer #1: No

Reviewer #2:**Yes:**Yuto MakinoYuto MakinoYuto MakinoYuto Makino

Reviewer #3: No

**Figure resubmission:**
---

## [Decision Letter · Decision Letter 3]

12 Mar 2026

Dear Dr. Rebelo Dal'Bello,

We are pleased to inform you that your manuscript 'A modular architecture for trial-by-trial learning of redundant muscle activity patterns in novel sensorimotor tasks' has been provisionally accepted for publication in PLOS Computational Biology.

Best regards,

Shlomi Haar, PhD

Academic Editor

PLOS Computational Biology

Andrea E. Martin

Section Editor

PLOS Computational Biology

Reviewer's Responses to Questions

**Comments to the Authors:**

Reviewer #1: I commend the authors for the substantial effort they have invested in revising this manuscript over multiple rounds of review. The revisions have been thorough and thoughtful, resulting in meaningful improvements to the manuscript. At this stage, I have no remaining concerns.

Reviewer #2: The authors have addressed my comments carefully and thoughtfully.

The manuscript provides a detailed and compelling account of how synergies expand toward the internal null space of the model and how adaptation to different perturbations is achieved within this framework. In particular, the model demonstrates strong explanatory power in accounting for behavioral patterns observed in virtual surgery experiments.

I believe this work provides a valuable foundation for future studies, including applications to BCI paradigms, more detailed descriptions of real motor behavior, and investigations into how such mechanisms may be implemented at the neural level.

I have no further concerns.

Reviewer #3: I would like to thank the authors for all their work in addressing the reviewers' comments.

I was able to run the simulation code without any issues. Thank you again!

**Have the authors made all data and (if applicable) computational code underlying the findings in their manuscript fully available?**

The PLOS Data policy requires authors to make all data and code underlying the findings described in their manuscript fully available without restriction, with rare exception (please refer to the Data Availability Statement in the manuscript PDF file). The data and code should be provided as part of the manuscript or its supporting information, or deposited to a public repository. For example, in addition to summary statistics, the data points behind means, medians and variance measures should be available. If there are restrictions on publicly sharing data or code —e.g. participant privacy or use of data from a third party—those must be specified.requires authors to make all data and code underlying the findings described in their manuscript fully available without restriction, with rare exception (please refer to the Data Availability Statement in the manuscript PDF file). The data and code should be provided as part of the manuscript or its supporting information, or deposited to a public repository. For example, in addition to summary statistics, the data points behind means, medians and variance measures should be available. If there are restrictions on publicly sharing data or code —e.g. participant privacy or use of data from a third party—those must be specified.requires authors to make all data and code underlying the findings described in their manuscript fully available without restriction, with rare exception (please refer to the Data Availability Statement in the manuscript PDF file). The data and code should be provided as part of the manuscript or its supporting information, or deposited to a public repository. For example, in addition to summary statistics, the data points behind means, medians and variance measures should be available. If there are restrictions on publicly sharing data or code —e.g. participant privacy or use of data from a third party—those must be specified.requires authors to make all data and code underlying the findings described in their manuscript fully available without restriction, with rare exception (please refer to the Data Availability Statement in the manuscript PDF file). The data and code should be provided as part of the manuscript or its supporting information, or deposited to a public repository. For example, in addition to summary statistics, the data points behind means, medians and variance measures should be available. If there are restrictions on publicly sharing data or code —e.g. participant privacy or use of data from a third party—those must be specified.

Reviewer #1: None

Reviewer #2: Yes

Reviewer #3: Yes

PLOS authors have the option to publish the peer review history of their article (what does this mean?). If published, this will include your full peer review and any attached files.). If published, this will include your full peer review and any attached files.). If published, this will include your full peer review and any attached files.). If published, this will include your full peer review and any attached files.

...

Reviewer #1: No

Reviewer #2: No

Reviewer #3: No

---

## [Editor Report · Acceptance letter]

PCOMPBIOL-D-25-00179R3

A modular architecture for trial-by-trial learning of redundant muscle activity patterns in novel sensorimotor tasks

Dear Dr Rebelo Dal'Bello,

I am pleased to inform you that your manuscript has been formally accepted for publication in PLOS Computational Biology. Your manuscript is now with our production department and you will be notified of the publication date in due course.

With kind regards,

Anita Estes
